# DynaConF:
# Dynamic Forecasting of Non-Stationary Time Series

**Siqi Liu**  *siqi.x.liu@borealisai.com*
*Borealis AI*

**Andreas Lehrmann**  *andreas.lehrmann@gmail.com*
*Borealis AI*

**Reviewed on OpenReview:** *https://openreview.net/forum?id=48pHFcg0YO*

## Abstract

Deep learning has shown impressive results in a variety of time series forecasting tasks, where modeling the conditional distribution of the future given the past is the essence. However, when this conditional distribution is non-stationary, it poses challenges for these models to learn consistently and to predict accurately. In this work, we propose a new method to model non-stationary conditional distributions over time by clearly decoupling stationary conditional distribution modeling from non-stationary dynamics modeling. Our method is based on a Bayesian dynamic model that can adapt to conditional distribution changes and a deep conditional distribution model that handles multivariate time series using a factorized output space. Our experimental results on synthetic and real-world datasets show that our model can adapt to non-stationary time series better than state-of-the-art deep learning solutions.

## 1 Introduction

Time series forecasting is a cornerstone of modern machine learning and has applications in a broad range of domains, such as operational processes (Salinas et al., 2020), energy (Lai et al., 2018), and transportation (Salinas et al., 2019). In recent years, models based on deep neural networks have shown particularly impressive results (Rangapuram et al., 2018; Salinas et al., 2019; 2020; Rasul et al., 2021b) and demonstrated the effectiveness of deep feature and latent state representations.

Despite this exciting progress, current time series forecasting methods often make the implicit assumption that the distribution of future observations conditioned on the input and past observations is time-invariant. In real-world applications this assumption can be violated, which poses serious practical challenges to a model's robustness and predictive power. The statistics literature studies several related concepts of (non-)stationarity for time series, with weak and strong stationarity being most widely used (Hamilton, 1994; Brockwell & Davis, 2009). Common to these variants of (non-)stationarity is that they are defined in terms of a stochastic process' joint or marginal distribution. For example, given a univariate time series $\{y_t \in \mathbb{R}\}_{t \in \mathbb{Z}}$ and any subset of time points $\{t_1, t_2, \ldots, t_k\}$, the time series is *strongly stationary* if $\forall \tau \in \mathbb{Z} : \mathrm{p}(y_{t_1}, y_{t_2}, \ldots, y_{t_k}) = \mathrm{p}(y_{t_1+\tau}, y_{t_2+\tau}, \ldots, y_{t_k+\tau})$.

While non-stationarity in a stochastic process' joint or marginal distribution is important and has been widely studied (Dickey & Fuller, 1979; Kwiatkowski et al., 1992; Hamilton, 1994; Kim et al., 2022), we argue that temporal *forecasting* relies more heavily on the properties of its *conditional* distribution $\mathrm{p}(\boldsymbol{y}_t | \boldsymbol{y}_{t-B:t-1}, \boldsymbol{x}_{t-B:t})$, where $\boldsymbol{y}_{t-B:t-1} = (\boldsymbol{y}_{t-B}, \boldsymbol{y}_{t-B+1}, \ldots, \boldsymbol{y}_{t-1})$, $\boldsymbol{y}_t$ is a real vector of the target variable, $B \in \mathbb{Z}_{>0}$ is the lookback window size and can be arbitrarily large, and $\boldsymbol{x}_t$ is a real vector containing auxiliary information. Most forecasting methods, from traditional approaches (e.g., Autoregressive Integrated Moving Average (ARIMA; (Box et al., 2015)), Generalized Autoregressive Conditional Heteroskedasticity (GARCH; (Engle, 1982; Bollerslev, 1986)), state-space models (SSMs; (Kalman, 1960))) to more recent models (e.g., recurrent

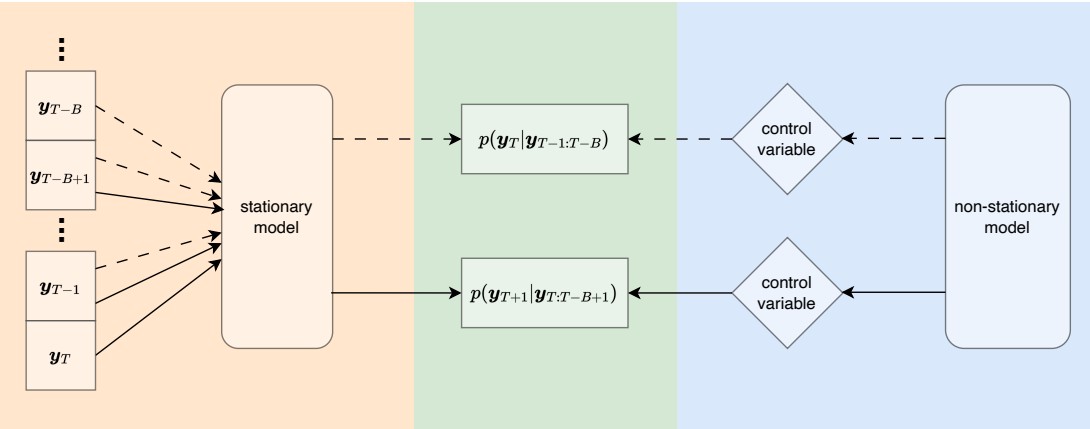

Figure 1: **Overview: Decoupled Model.** DynaConF decouples the stationary (time-invariant) modeling (left) and the non-stationary (time-variant) modeling (right) of the conditional distribution $p(\boldsymbol{y}_t|\boldsymbol{y}_{t-1:t-B})$ across time $t$ (middle). The dashed arrows indicate the information flow at time $T-1$ when predicting $\boldsymbol{y}_T$, while the solid arrows indicate the information flow at time $T$ when predicting $\boldsymbol{y}_{T+1}$.

neural networks (RNNs; (Hochreiter & Schmidhuber, 1997; Salinas et al., 2020)), temporal convolutional networks (TCNs; (Bai et al., 2018)), Transformers (Vaswani et al., 2017; Rasul et al., 2021b; Wu et al., 2021; Nie et al., 2022; Drouin et al., 2022), Neural Processes (NPs; (Garnelo et al., 2018a;b))), rely on this conditional distribution for predictions, but many of them implicitly assume its stationarity by using time-invariant model parameters. While a model with stationary conditional distribution can still handle non-stationarity in the joint or marginal distribution, such as seasonality and trend, by conditioning on extra features in $\boldsymbol{x}_t$, such as day of the week, the conditional distribution itself may also change due to (1) unobserved causes and/or (2) new causes. For example, the daily number of posts from each user on a social media platform is unlikely to be robustly predictable from historical data, even with input features like day of the week, because (1) user activity is often affected by events that are not reflected in the observable data (e.g., illness); and (2) events that have not been seen before may occur (e.g., a new functionality being added to the platform) and change the functional pattern of the input-output relation between the conditioning and target variables in an unpredictable way. How to deal with these changes in the *conditional* distribution is the main focus of this work.

Autoregressive (AR) models, TCNs, and Transformers model $p(\boldsymbol{y}_t|\boldsymbol{y}_{t-B:t-1}, \boldsymbol{x}_{t-B:t})$ with time-invariant parameters (e.g., the weights and biases in the neural network) and therefore assume stationarity in $p(\boldsymbol{y}_t|\boldsymbol{y}_{t-B:t-1}, \boldsymbol{x}_{t-B:t})$. In contrast, SSMs, RNNs, and NPs model $p(\boldsymbol{y}_t|\boldsymbol{y}_{1:t-1}, \boldsymbol{x}_{1:t})$ with time-invariant parameters as well, but they have a growing number of conditioning variables (note the time range $1:t$) and therefore can potentially model different conditional distributions $p(\boldsymbol{y}_t|\boldsymbol{y}_{t-B:t-1}, \boldsymbol{x}_{t-B:t})$ at different time points $t$. However, these models need to achieve two goals using the same time-invariant structure: (1) modeling $p(\boldsymbol{y}_t|\boldsymbol{y}_{t-B:t-1}, \boldsymbol{x}_{t-B:t})$; and (2) modeling its changes over time. Because they do not incorporate explicit inductive biases for changes in $p(\boldsymbol{y}_t|\boldsymbol{y}_{t-B:t-1}, \boldsymbol{x}_{t-B:t})$, they either cannot learn different (seemingly inconsistent) relations between the conditioning and target variables (if the model capacity is limited) or tend to memorize the training data and are not able to generalize to new changes at test time.

In this work, we take a different approach to dealing with non-stationary conditional distributions in time series. The core of our model, called DynaConF[1], is a clean decoupling of the time-variant (non-stationary) and the time-invariant (stationary) part of the distribution. The time-invariant part models a stationary conditional distribution, given a control variable, while the time-variant part focuses on modeling the changes in this conditional distribution over time through the control variable (Figure 1). Using this separation, we build a flexible model for the conditional distribution of the time series and make efficient inferences about its changes over time. At test time, our model takes both the uncertainty from the conditional distribution itself and non-stationary effects into account when making predictions and can adapt to *unseen* changes in the conditional distribution over time in an *online* manner.

---

[1] https://github.com/BorealisAI/dcf

## 2 Related Work

Time series forecasting has a rich history (Hamilton, 1994; Box et al., 2015), with many recent advances due to deep neural networks (Lim & Zohren, 2021). The following paragraphs discuss different approaches to non-stationary time series modeling and their relation to our work.

**Non-Stationary Marginal Distribution.** There are three common ways of dealing with non-stationarity in the marginal distribution: (1) **Data transformation.** In ARIMA models, taking the difference of the time series over time can remove trend and seasonality. More advanced approaches based on exponential smoothing (Holt, 2004) or seasonal-trend decomposition (Cleveland et al., 1990) have also been combined with deep neural networks, achieving promising performance (Smyl, 2020; Bandara et al., 2020). More recently, Kim et al. (2022) propose to use reversible normalization/denormalization on the input/output of the time series model to account for (marginal) distribution shifts over time. (2) **Inductive bias.** As an alternative to data transformations, the underlying ideas of exponential smoothing and decomposition can also be incorporated into deep learning models as inductive biases, which enables end-to-end handling of seasonality/trend (Lai et al., 2018; Oreshkin et al., 2019; Wu et al., 2021; Woo et al., 2022). Similarly, in models based on Gaussian processes, the inductive biases can be added as specific (e.g., periodic or linear) kernel choices (Corani et al., 2021), and in models based on temporal matrix factorization as part of the loss function (Chen et al., 2022). (3) **Conditioning information.** Adding features such as relative time (e.g., day of the week) and absolute time to the model input as conditioning information is commonly used in deep probabilistic forecasting models (Rangapuram et al., 2018; Salinas et al., 2019; 2020; Rasul et al., 2021b;a). In our work, we focus on proper handling of changes in the *conditional* distribution. To deal with marginal distribution shifts, we simply add conditioning information as in approach (3), although we could potentially utilize approach (1) and (2) as well.

**Non-Stationary Conditional Distribution.** State-space models (Fraccaro et al., 2017; Rangapuram et al., 2018; de Bézenac et al., 2020; Tang & Matteson, 2021; Klushyn et al., 2021) and recurrent neural networks (Salinas et al., 2019; 2020; Rasul et al., 2021b;a) are among the most popular choices to model temporal dependencies in time series. When these models are applied to, and therefore conditioned on, the entire history of the time series, they can theoretically "memorize" different conditional distributions at different points in time. However, for these models to generalize and adapt to new changes in the future, it is critical to have appropriate inductive biases built into the model. A common approach is to allow state-space models to switch between a discrete set of dynamics, which can be learned from training data (Kurle et al., 2020; Ansari et al., 2021). However, the model cannot adapt to continuous changes or generalize to new changes that have not been observed in the training data. In contrast, our model has explicit inductive biases to account for both continuous and discontinuous changes, and to adapt to new changes. Yanchenko & Mukherjee (2020) proposed a nonlinear state-space model with a time-varying transition matrix for the latent state. In contrast, our model is a *decoupled deep* model separating the time-invariant conditional distribution modeling using a *deep* encoder and the non-stationary dynamics modeling.

**Observation Model.** The expressivity and flexibility of the observation model is a topic that is especially relevant in case of multivariate time series. Different observation models have been employed in time series models, including low-rank covariance structures (Salinas et al., 2019), auto-encoders (Fraccaro et al., 2017; Nguyen & Quanz, 2021), normalizing flows (de Bézenac et al., 2020; Rasul et al., 2021b), determinantal point processes (Le Guen & Thome, 2020), denoising diffusion models (Rasul et al., 2021a; Tashiro et al., 2021; Alcaraz & Strodthoff, 2022), and probabilistic circuits (Yu et al., 2021). In this work, we prioritize scalability by assuming a simple conditional independence structure in the output space and consider more expressive observation models as orthogonal to our work.

**Online Approaches.** Continual learning (Kirkpatrick et al., 2017; Nguyen et al., 2018; Kurle et al., 2019; Parisi et al., 2019; De Lange et al., 2021; Gupta et al., 2021) also addresses (conditional) distribution changes in an online manner, but usually in a multi-task supervised learning setting and not in time series. In this work, our focus is on conditional distribution changes in time series, and the conditional distribution can change either continuously or discontinuously in time.

## 3 Method

We study the problem of modeling and forecasting time series with changes in the conditional distribution $p(\boldsymbol{y}_t|\boldsymbol{y}_{t-B:t-1},\boldsymbol{x}_{t-B:t})$ over time $t$, where $\boldsymbol{y}_t \in \mathbb{R}^{D_y}$ is the target time series, and $\boldsymbol{x}_t \in \mathbb{R}^{D_x}$ is an input containing contextual information. We make the following assumption:

**Assumption 1** $\boldsymbol{y}_t$ *only depends on a bounded history of* $\boldsymbol{y}$ *and* $\boldsymbol{x}$ *for all* $t$*. That is, there exists* $B \in \mathbb{Z}_{>0}$ *such that for all* $t$*,* $p(\boldsymbol{y}_t|\boldsymbol{y}_{<t},\boldsymbol{x}_{\leq t}) = p(\boldsymbol{y}_t|\boldsymbol{y}_{t-B:t-1},\boldsymbol{x}_{t-B:t})$*.*

Although we assume that $\boldsymbol{y}_t$ only depends on the history up to $B$ time steps, its conditional distribution can change over time based on information beyond $B$ steps. Assumption 1 is not particularly restrictive in practice, since we usually have a finite amount of training data while $B$ can be arbitrarily large (although usually not needed). Given historical data $\{(\boldsymbol{x}_t,\boldsymbol{y}_t)\}_{t=1}^{T}$, our task is to fit a model to the data and use it to forecast $\{\boldsymbol{y}_t\}_{t=T+1}^{T+H}, \{\boldsymbol{y}_t\}_{t=T+H+1}^{T+H+H}, \ldots$, continually with a horizon and step size $H$, given $\{\boldsymbol{x}_t\}_{t=T+1}^{T+H}, \{\boldsymbol{x}_t\}_{t=T+H+1}^{T+H+H}, \ldots$. At time $T$, $\boldsymbol{x}_t$, for $t \leq T$, may contain any information known at time $t$, while $\boldsymbol{x}_t$, for $t > T$, can only contain information known in advance, such as day of the week. We do not distinguish between these two cases in notation for simplicity.

### 3.1 Decoupled Model

We assume that the distribution $p(\boldsymbol{y}_t|\boldsymbol{y}_{t-B:t-1},\boldsymbol{x}_{t-B:t})$ is from a distribution family parametrizable by $\boldsymbol{\theta}_t \in \mathbb{R}^{D_\theta}$. Concretely, we assume a normal distribution $\mathcal{N}(\boldsymbol{\mu}_t, \boldsymbol{\Sigma}_t)$, so $\boldsymbol{\theta}_t := (\boldsymbol{\mu}_t, \boldsymbol{\Sigma}_t)$. Our decoupled model is not restricted to this assumption, but as we will see in Section 3.5 it allows for more efficient inference. Furthermore, we assume

$$\boldsymbol{\theta}_t = f_{\boldsymbol{\psi},\boldsymbol{\phi}_t}(\boldsymbol{y}_{t-B:t-1},\boldsymbol{x}_{t-B:t}), \tag{1}$$

so $f : \mathbb{R}^{B \times D_y} \times \mathbb{R}^{(B+1) \times D_x} \to \mathbb{R}^{D_\theta}$ models the conditional distribution, with its own static parameters denoted collectively as $\boldsymbol{\psi}$, and is modulated by a dynamic control variable $\boldsymbol{\phi}_t \in \mathbb{R}^{D_\phi}$, which can change over time according to a dynamic process defined in Section 3.3. A property guaranteed in our model is that if $\boldsymbol{\phi}_t$ stays the same across time, then the conditional distribution $p(\boldsymbol{y}_t|\boldsymbol{y}_{t-B:t-1},\boldsymbol{x}_{t-B:t})$ stays the same as well.

Our key idea is to separate the time-variant part of the model from the time-invariant part, instead of allowing all components to vary across time. This simplifies probabilistic inference and improves the generalization capabilities of the learned model by allowing the time-invariant part to learn from time-invariant input-output relations with time-variant modulations.

### 3.2 Conditional Distribution at One Time Point

First we describe how we model the conditional distribution $p(\boldsymbol{y}_t|\boldsymbol{y}_{t-B:t-1},\boldsymbol{x}_{t-B:t})$ at each time point $t$ without accounting for non-stationary effects (Figure 1, left). We use a neural network $g$ to encode the historical and contextual information into a vector $\boldsymbol{h}_t \in \mathbb{R}^{D_h}$ as $\boldsymbol{h}_t = g(\boldsymbol{y}_{t-B:t-1},\boldsymbol{x}_{t-B:t})$. For example, $g$ could be a multi-layer perceptron (MLP) or a recurrent neural network (RNN). The parameters of $g$ are time-invariant and included in $\boldsymbol{\psi}$ (Eq. 1).

We note that a key distinction between our model's use of an RNN and a typical deep time series model using an RNN is that the latter keeps unrolling the RNN over time to model the dynamics of the time series. In contrast, we unroll the RNN for $B + 1$ steps to summarize $(\boldsymbol{y}_{t-B:t-1},\boldsymbol{x}_{t-B:t})$ in the exact same way at each time point $t$, i.e., we apply it in a *time-invariant* manner.

We construct the distribution of $\boldsymbol{y}_t$ such that each dimension $i$ of $\boldsymbol{y}_t$, denoted as $y_{t,i}$, is conditionally independent of the others given $\boldsymbol{h}_t$; this helps the learning and inference algorithms to scale better with the dimensionality of $\boldsymbol{y}_t$. First we transform $\boldsymbol{h}_t$ into $D_y$ vectors $\boldsymbol{z}_{t,i} \in \mathbb{R}^{E}$ as

$$\boldsymbol{z}_{t,i} = \tanh(\boldsymbol{W}_{z,i}\boldsymbol{h}_t + \boldsymbol{b}_{z,i}), \quad \forall i = 1, \ldots, D_y \tag{2}$$

where $\boldsymbol{W}_{z,i} \in \mathbb{R}^{E \times D_h}$ and $\boldsymbol{b}_{z,i} \in \mathbb{R}^{E}$, so $\boldsymbol{z}_{t,i}$ corresponds to $y_{t,i}$. Then, from $\boldsymbol{z}_{t,i}$, we obtain the distribution parameters $\boldsymbol{\theta}_{t,i}$ of $y_{t,i}$. Specifically, we assume a normal distribution with a diagonal covariance for $\boldsymbol{y}_t$, so

$y_{t,i} \sim \mathcal{N}(\mu_{t,i}, \sigma_{t,i}^2)$ and $\boldsymbol{\theta}_{t,i} := (\mu_{t,i}, \sigma_{t,i}^2)$, with

$$\mu_{t,i} = \boldsymbol{w}_{\mu,i}^T \boldsymbol{z}_{t,i} + b_{\mu,i}, \quad \sigma_{t,i} = \exp(\boldsymbol{w}_{\sigma,i}^T \boldsymbol{z}_{t,i} + b_{\sigma,i}), \tag{3}$$

where $\boldsymbol{w}_{\mu,i} \in \mathbb{R}^E$ and $\boldsymbol{w}_{\sigma,i} \in \mathbb{R}^E$. We use $\boldsymbol{w}_\mu \in \mathbb{R}^{E \cdot D_y}$ and $\boldsymbol{b}_\mu \in \mathbb{R}^{D_y}$ to denote the concatenation of $\boldsymbol{w}_{\mu,i}^T$ and $b_{\mu,i}$ along $i$, and similarly $\boldsymbol{w}_\sigma, \boldsymbol{b}_\sigma, \boldsymbol{z}_t, \boldsymbol{\mu}_t, \boldsymbol{\sigma}_t$.

### 3.3 Conditional Distributions Across Time Points

We have explained how we model the conditional distribution $\mathrm{p}(\boldsymbol{y}_t | \boldsymbol{y}_{t-B:t-1}, \boldsymbol{x}_{t-B:t})$ at each time point $t$. To model changes in the conditional distribution over time, we first specify which parameters to include in the control variable $\boldsymbol{\phi}_t \in \mathbb{R}^{D_\phi}$, which changes over time and modulates the conditional distribution (Figure 1, right).

We choose $\boldsymbol{\phi}_t := \boldsymbol{w}_\mu$. Recall that $\boldsymbol{w}_\mu$ transforms $\boldsymbol{z}_t$ into the mean $\boldsymbol{\mu}_t$ of $\boldsymbol{y}_t$, where $\boldsymbol{z}_t$ is a summary of the information in the conditioning variables $(\boldsymbol{y}_{t-B:t-1}, \boldsymbol{x}_{t-B:t})$. By allowing $\boldsymbol{w}_\mu$ to be different at each time point $t$, the conditional mean of $\boldsymbol{y}_t$, $\mathrm{E}[\boldsymbol{y}_t | \boldsymbol{y}_{t-B:t-1}, \boldsymbol{x}_{t-B:t}]$, can change accordingly. We could allow $\boldsymbol{w}_\sigma$ to change over time as well, effectively modeling a time-variant conditional variance, but focusing on $\boldsymbol{w}_\mu$ reduces the dimensionality of $\boldsymbol{\phi}_t$ and enables more efficient inference utilizing Rao-Blackwellization (Section 3.5).

We propose to decompose $\boldsymbol{\phi}_t$ into a dynamic stochastic process $\boldsymbol{\chi}_t \in \mathbb{R}^{D_\phi}$ and a static vector $\boldsymbol{b}_\phi \in \mathbb{R}^{D_\phi}$:

$$\boldsymbol{\phi}_t = \boldsymbol{\chi}_t + \boldsymbol{b}_\phi. \tag{4}$$

The intuition is that $\boldsymbol{b}_\phi$ captures the global information of $\boldsymbol{\phi}_t$ and acts as a baseline, while $\boldsymbol{\chi}_t$ captures the time-variant changes of $\boldsymbol{\phi}_t$ relative to $\boldsymbol{b}_\phi$.

We assume that $\boldsymbol{\chi}_t$ follows the generative process

$$\begin{aligned} \pi_t &\sim \mathcal{B}(\lambda); & \boldsymbol{\chi}_t &\sim \mathcal{N}(\mathbf{0}, \boldsymbol{\Sigma}_\chi), \text{ if } \pi_t = 0; \\ \boldsymbol{\epsilon}_t &\sim \mathcal{N}(\mathbf{0}, \boldsymbol{\Sigma}_\epsilon); & \boldsymbol{\chi}_t &= \boldsymbol{\chi}_{t-1} + \boldsymbol{\epsilon}_t, \text{ if } \pi_t = 1. \end{aligned} \tag{5}$$

$\mathcal{B}$ and $\mathcal{N}$ denote the Bernoulli and normal distributions, respectively. $\pi_t \in \{0, 1\}$ is a binary random variable that decides whether generating the current $\boldsymbol{\chi}_t$ as a new sample drawn from a global distribution $\mathcal{N}(\mathbf{0}, \boldsymbol{\Sigma}_\chi)$, or as a continuation from the previous $\boldsymbol{\chi}_{t-1}$ following a simple stochastic process in the form of a random walk. The intention is to allow $\boldsymbol{\chi}_t$ to change both continuously (when $\pi_t = 1$) through the random walk and discontinuously (when $\pi_t = 0$) through the global distribution, which captures the variety of possible changes of $\boldsymbol{\chi}_t$ in its parameter $\boldsymbol{\Sigma}_\chi$.

We let $\boldsymbol{\chi}_t$ start at $t = B$ instead of $t = 0$, since it controls the *conditional* distribution of the time series, which requires at least $B$ observations in the past. For the initial $\boldsymbol{\chi}_B$, we assume generation from $\mathcal{N}(\mathbf{0}, \boldsymbol{\Sigma}_\chi)$ as well. Our intention is that $\mathcal{N}(\mathbf{0}, \boldsymbol{\Sigma}_\chi)$ should be the distribution to generate new $\boldsymbol{\chi}_t$ whenever there is a drastic change in the conditional distribution of $\boldsymbol{y}_t$, so at $t = B$ it is natural to use that distribution.

Recall that $\boldsymbol{\chi}_t \in \mathbb{R}^{D_\phi}$, with $D_\phi = E \cdot D_y$. We propose to separate $\boldsymbol{\chi}_t$ along the dimensions of $\boldsymbol{y}_t$ into $D_y$ groups. For each $i = 1, \ldots, D_y$, we define $\boldsymbol{\chi}_{t,i} \in \mathbb{R}^E$ as in Eq. 5. The final $\boldsymbol{\chi}_t$ is the concatenation of $\boldsymbol{\chi}_{t,i}$ for all $i$. The intuition is to allow the group of components of $\boldsymbol{\chi}_t$ modulating each dimension of $\boldsymbol{y}_t$ to change independently of the others, corresponding to the conditional independence assumption we made in Section 3.2. We can thus sample a subset of dimensions of $\boldsymbol{y}_t$ in each iteration during training to scale to high-dimensional time series. Our full model is shown in Figure 2.

### 3.4 Learning

All parameters of the conditional distribution model and the prior, i.e., $\{\boldsymbol{\psi}, \boldsymbol{b}_\phi, \lambda, \boldsymbol{\Sigma}_\chi, \boldsymbol{\Sigma}_\epsilon\}$, are learned by fitting the model to the historical training data $\{(\boldsymbol{y}_t, \boldsymbol{x}_t)\}_{t=1}^T$. We train our model by maximizing the marginal log-likelihood, where the latent variables $\boldsymbol{\chi}_t$ are marginalized out. Given a trajectory of $\boldsymbol{\chi}_{B:T}$, the conditional log-likelihood is

$$\log \mathrm{p}(\boldsymbol{y}_{B+1:T} | \boldsymbol{y}_{1:B}, \boldsymbol{x}_{1:T}, \boldsymbol{\chi}_{B:T}) = \sum_{t=B+1}^T \log \mathrm{p}(\boldsymbol{y}_t | \boldsymbol{y}_{t-B:t-1}, \boldsymbol{x}_{t-B:t}, \boldsymbol{\chi}_t). \tag{6}$$

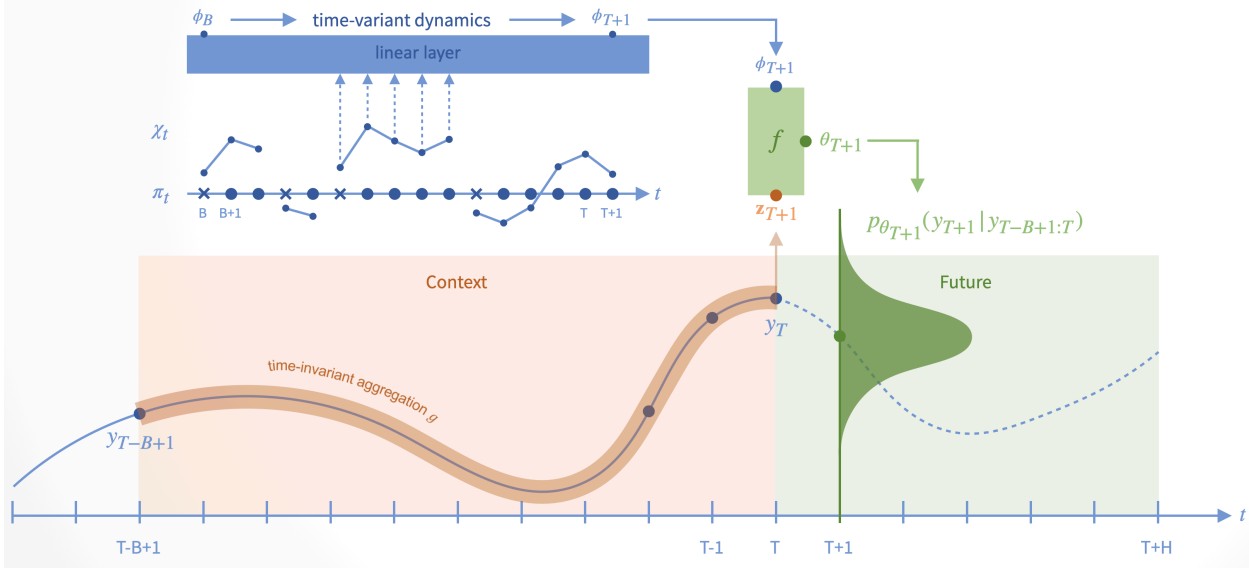

Figure 2: **Architecture.** DynaConF is built on the principle of a clean decoupling of stationary conditional distribution modeling (red) and non-stationary dynamics modeling (blue). We predict the parameters ($\theta$) of the conditional distribution (green) by aggregating time-invariant local context ($z$) and modulating this context with time-variant global dynamics ($\phi$) driven by a random walk ($\chi$) with Bernoulli restarts ($\pi$).

Marginalizing out $\boldsymbol{\chi}_t$ gives the log-likelihood objective

$$\log \mathrm{p}(\boldsymbol{y}_{B+1:T}|\boldsymbol{y}_{1:B}, \boldsymbol{x}_{1:T}) = \log \int \mathrm{p}(\boldsymbol{y}_{B+1:T}|\boldsymbol{y}_{1:B}, \boldsymbol{x}_{1:T}, \boldsymbol{\chi}_{B:T})\mathrm{p}(\boldsymbol{\chi}_{B:T})\mathrm{d}\boldsymbol{\chi}_{B:T}. \tag{7}$$

Since the integral is intractable, we instead introduce a variational distribution $\mathrm{q}(\boldsymbol{\chi}_{B:T})$ and maximize the following variational lower-bound $\mathcal{L}$ of the log-likelihood in Eq. 7:

$$\mathcal{L} := \mathrm{E}_{\mathrm{q}}[\log \mathrm{p}(\boldsymbol{y}_{B+1:T}|\boldsymbol{y}_{1:B}, \boldsymbol{x}_{1:T}, \boldsymbol{\chi}_{B:T})] + \mathrm{E}_{\mathrm{q}}\left[\log \frac{\mathrm{p}(\boldsymbol{\chi}_{B:T})}{\mathrm{q}(\boldsymbol{\chi}_{B:T})}\right] \le \log \mathrm{p}(\boldsymbol{y}_{B+1:T}|\boldsymbol{y}_{1:B}, \boldsymbol{x}_{1:T}). \tag{8}$$

Based on the conditional independence structure of the prior process, we construct the variational distribution $\mathrm{q}(\boldsymbol{\chi}_{B:T})$ similar to an autoregressive process. However, we assume a simple normal distribution at each time step for efficient sampling and back-propagation. First, we define $\mathrm{q}(\boldsymbol{\chi}_B)$ as a normal distribution. Then, conditioning on the previous $\boldsymbol{\chi}_{t-1}$, we recursively define

$$\mathrm{q}(\boldsymbol{\chi}_t|\boldsymbol{\chi}_{t-1}) = \mathcal{N}(\boldsymbol{a}_t \odot \boldsymbol{\chi}_{t-1} + (1 - \boldsymbol{a}_t) \odot \boldsymbol{m}_t, \mathrm{diag}(\boldsymbol{s}_t^2)), \quad \forall t = B+1, \ldots, T, \tag{9}$$

where $\boldsymbol{a}_t, \boldsymbol{m}_t, \boldsymbol{s}_t \in \mathbb{R}^{D_\phi}$ are variational parameters. Intuitively, $\boldsymbol{a}_t$ is a gate that chooses between continuing from the previous $\boldsymbol{\chi}_{t-1}$, with noise $\mathcal{N}(0, \mathrm{diag}(\boldsymbol{s}_t^2))$, and using a new distribution $\mathcal{N}(\boldsymbol{m}_t, \mathrm{diag}(\boldsymbol{s}_t^2))$.

We note that both terms in Eq. 8 factorize over time $t = B+1, \ldots, T$,

$$
\begin{aligned}
\mathcal{L} &= \mathrm{E}_{\mathrm{q}(\boldsymbol{\chi}_{B:T})}\left[\sum_{t=B+1}^{T} \log \mathrm{p}(\boldsymbol{y}_t|\boldsymbol{y}_{t-B:t-1}, \boldsymbol{x}_{t-B:t}, \boldsymbol{\chi}_t)\right] + \mathrm{E}_{\mathrm{q}(\boldsymbol{\chi}_{B:T})}\left[\sum_{t=B}^{T} \log \frac{\mathrm{p}(\boldsymbol{\chi}_t|\boldsymbol{\chi}_{t-1})}{\mathrm{q}(\boldsymbol{\chi}_t|\boldsymbol{\chi}_{t-1})}\right] \\
&= \sum_{t=B+1}^{T} \mathrm{E}_{\mathrm{q}(\boldsymbol{\chi}_t)}\left[\log \mathrm{p}(\boldsymbol{y}_t|\boldsymbol{y}_{t-B:t-1}, \boldsymbol{x}_{t-B:t}, \boldsymbol{\chi}_t)\right] + \sum_{t=B}^{T} \mathrm{E}_{\mathrm{q}(\boldsymbol{\chi}_{t-1:t})}\left[\log \frac{\mathrm{p}(\boldsymbol{\chi}_t|\boldsymbol{\chi}_{t-1})}{\mathrm{q}(\boldsymbol{\chi}_t|\boldsymbol{\chi}_{t-1})}\right],
\end{aligned}
\tag{10}
$$

where, to simplify notation, we define $\mathrm{p}(\boldsymbol{\chi}_t|\boldsymbol{\chi}_{t-1})$ at $t = B$ to be $\mathrm{p}(\boldsymbol{\chi}_B)$, and similarly for q. The expectations in this equation can be evaluated by Monte-Carlo sampling from $\mathrm{q}(\boldsymbol{\chi})$ with the reparameterization trick (Kingma & Welling, 2013) for back-propagation.

In practice, sequential sampling in the autoregressive posterior could be inefficient because the sampling cannot be parallelized. We further develop a generalized posterior by replacing the autoregressive chain with multiple moving-average blocks combined with autoregressive dependencies between consecutive blocks, where sampling within each block can be carried out in parallel. Specifically, for the $i$-th time block, $t \in (b_i, b_{i+1}]$, where $b_1 = B, b_{K+1} = T, b_i < b_{i+1}$, out of $K$ blocks, we sample

$$\boldsymbol{\delta}_t \sim \mathcal{N}((1 - \boldsymbol{a}_t) \odot \boldsymbol{m}_t, \operatorname{diag}(\boldsymbol{s}_t^2)) \tag{11}$$

in parallel across $t$ and compute

$$\boldsymbol{\chi}_t = \left(\prod_{v \in (b_i, t]} \boldsymbol{a}_v\right) \odot \boldsymbol{\chi}_{b_i} + \sum_{u \in (b_i, t]} \left(\prod_{v \in (u, t]} \boldsymbol{a}_v\right) \odot \boldsymbol{\delta}_u. \tag{12}$$

This generalizes the autoregressive posterior and is the form used in our experiments.

Utilizing our modeling assumptions from Section 3.2 and Section 3.3, we develop an SGD-based optimization protocol suitable for large datasets. Specifically, we alternate between optimizing the conditional distribution model and the prior and posterior of the dynamic model with different sampling strategies to accommodate high dimensionality and long time spans. See Appendix A for more details about our optimization process.

## 3.5 Forecasting

At test time, we are given past observations $\boldsymbol{y}_{1:T}$ as well as input features $\boldsymbol{x}_{1:T+H}$, including $H$ future steps, to infer the conditional distribution $\mathrm{p}(\boldsymbol{y}_{T+1:T+H} | \boldsymbol{y}_{1:T}, \boldsymbol{x}_{1:T+H})$. We note again that $\boldsymbol{x}_{T+1:T+H}$ only contains information known ahead, such as day of the week. Based on our modeling assumptions, the conditional distribution can be computed as

$$
\begin{aligned}
&\mathrm{p}(\boldsymbol{y}_{T+1:T+H} | \boldsymbol{y}_{1:T}, \boldsymbol{x}_{1:T+H}) \\
&= \int \mathrm{p}(\boldsymbol{y}_{T+1:T+H} | \boldsymbol{y}_{T+1-B:T}, \boldsymbol{x}_{T+1-B:T+H}, \boldsymbol{\chi}_{T+1:T+H}) \mathrm{p}(\boldsymbol{\chi}_{T+1:T+H} | \boldsymbol{y}_{1:T}, \boldsymbol{x}_{1:T}) \mathrm{d}\boldsymbol{\chi}_{T+1:T+H}.
\end{aligned} \tag{13}
$$

The first factor in the integrand can be computed recursively via step-by-step predictions based on our conditional distribution model given $\boldsymbol{\chi}_{T+1:T+H}$,

$$\mathrm{p}(\boldsymbol{y}_{T+1:T+H} | \boldsymbol{y}_{T+1-B:T}, \boldsymbol{x}_{T+1-B:T+H}, \boldsymbol{\chi}_{T+1:T+H}) = \prod_{t=T+1}^{T+H} \mathrm{p}(\boldsymbol{y}_t | \boldsymbol{y}_{t-B:t-1}, \boldsymbol{x}_{t-B:t}, \boldsymbol{\chi}_t). \tag{14}$$

The second factor in the integrand can be further factorized into

$$\mathrm{p}(\boldsymbol{\chi}_{T+1:T+H} | \boldsymbol{y}_{1:T}, \boldsymbol{x}_{1:T}) = \int \mathrm{p}(\boldsymbol{\chi}_{T+1:T+H} | \boldsymbol{\chi}_T) \mathrm{p}(\boldsymbol{\chi}_T | \boldsymbol{y}_{1:T}, \boldsymbol{x}_{1:T}) \mathrm{d}\boldsymbol{\chi}_T. \tag{15}$$

We use Rao-Blackwellized particle filters (Doucet et al., 2000) to infer $\mathrm{p}(\boldsymbol{\chi}_T | \boldsymbol{y}_{1:T}, \boldsymbol{x}_{1:T})$, so our model can keep adapting to new changes in an online manner, where the Rao-Blackwellization is possible because of our distribution assumptions made in the previous sections. We jointly infer $\boldsymbol{\pi}_t$ and $\boldsymbol{\chi}_t$, with particles representing $\boldsymbol{\pi}_t$ and closed-form inference of $\boldsymbol{\chi}_t$. Once we have obtained the posterior samples of $\mathrm{p}(\boldsymbol{\chi}_T | \boldsymbol{y}_{1:T}, \boldsymbol{x}_{1:T})$, we use the prior model to sample trajectories of $\boldsymbol{\chi}_{T+1:T+H}$ conditioned on the samples of $\boldsymbol{\chi}_T$. With the sample trajectories of $\mathrm{p}(\boldsymbol{\chi}_{T+1:T+H} | \boldsymbol{y}_{1:T}, \boldsymbol{x}_{1:T})$, we sample the trajectories of $\mathrm{p}(\boldsymbol{y}_{T+1:T+H} | \boldsymbol{y}_{T+1-B:T}, \boldsymbol{x}_{T+1-B:T+H}, \boldsymbol{\chi}_{T+1:T+H})$ using the aforementioned step-by-step predictions with our conditional distribution model.

## 4 Experiments

We compare our approach with 2 univariate and 9 multivariate time series models on synthetic (Section 4.1) and real-world (Section 4.2 and 4.3) datasets; see Table 1 for an overview, including references to the relevant literature and implementations. We note that DeepVAR is also called Vec-LSTM in previous works (Salinas et al., 2019; Rasul et al., 2021b). For our own model we consider two variants: an ablated model without the

Table 1: **Baseline Models.**

| Method | S | R | M | P | N | Implementation |
|---|:---:|:---:|:---:|:---:|:---:|---|
| DeepAR (Salinas et al., 2020) | ✓ | | | ✓ | | GluonTS[2](Alexandrov et al., 2020) |
| DeepSSM (Rangapuram et al., 2018) | ✓ | | | ✓ | | GluonTS |
| TransformerMAF (Rasul et al., 2021b) | ✓ | ✓ | ✓ | ✓ | | PyTorchTS[3] |
| DeepVAR (Salinas et al., 2019) | ✓ | ✓ | ✓ | ✓ | | GluonTS |
| GP-Copula (Salinas et al., 2019) | | ✓ | ✓ | ✓ | | GluonTS |
| LSTM-MAF (Rasul et al., 2021b) | | ✓ | ✓ | ✓ | | PyTorchTS |
| TimeGrad (Rasul et al., 2021a) | | ✓ | ✓ | ✓ | | PyTorchTS |
| TACTiS (Drouin et al., 2022) | | ✓ | ✓ | ✓ | | Authors'[4] |
| NS Tranformer (Liu et al., 2022) | | ✓ | ✓ | | ✓ | TSlib[5](Wu et al., 2022) |
| DeepTime (Woo et al., 2023) | | ✓ | ✓ | | ✓ | Authors'[6] |
| Koopa (Liu et al., 2023) | | ✓ | ✓ | | ✓ | TSlib |

[S = Synthetic; R = Real-World; M = Multivariate; P = Probabilistic; N = Non-Stationary]

dynamic updates to the conditional distribution described in Section 3.3 (StatiConF); and our full model including those contributions (DynaConF). In both cases we experiment with different encoder architectures. For synthetic data, we use either a two-layer MLP with 32 hidden units (*–MLP) or a point-wise linear + tanh mapping (*–PP) as the encoder. For real-world data, we use an LSTM as the encoder.

## 4.1 Experiments on Synthetic Data

**Datasets** For our experiments on synthetic data we simulate four conditionally non-stationary stochastic processes for $T = 2500$ time steps, where we use the first 1000 steps as training data, the next 500 steps as validation data, and the remaining 1000 steps as test data: (*AR(1)–Flip*) We simulate an AR(1) process, $y_t = w_t y_{t-1} + \epsilon_t, \epsilon_t \sim \mathcal{N}(0,1)$, but resample its coefficient $w_t$ from a uniform categorical distribution over $\{-0.5, +0.5\}$ every 100 steps to introduce non-stationarity. (*AR(1)–Dynamic*) We simulate the same process as above but now resample $w_t$ from a continuous uniform distribution $\mathcal{U}(-1, 1)$ every 100 steps. (*AR(1)–Sin*) We simulate the same process as above but now resample $w_t$ according to $w_t = \sin(2\pi t/T)$. Different from the two processes above, this process has a continuously changing non-stationary conditional distribution with its own time-dependent dynamics. (*VAR(1)–Dynamic*) This process can be viewed as a multivariate generalization of AR(1)–Dynamic and is used in our comparisons with multivariate baselines. We use a four-dimensional VAR process with an identity noise matrix. Similar to the univariate case, we resample the entries of the coefficient matrix from a continuous uniform distribution $\mathcal{U}(-0.8, 0.8)$ every 250 steps. In addition, we ensure the stability of the resulting process by computing the eigenvalues of the coefficient matrix and discard it if its largest absolute eigenvalue is greater than 1.

**Experimental Setup** For univariate data (AR(1)–Flip/Sin/Dynamic), we compare our approach with the univariate baselines DeepAR and DeepSSM. For multivariate data (VAR(1)–Dynamic), most baselines are redundant because their focus is on better observation models, while the underlying temporal backbone is similar. Since our synthetic observation distributions are simple, we compare with two model families that differ in their temporal backbone: DeepVAR (RNN backbone) and TransformerMAF (Transformer backbone). We use a lookback window size of 200 to give the models access to the information needed to infer the current parameter of the true conditional distribution. We tried increasing the window size to 500 on VAR(1)–Dynamic but did not see performance improvements. We also removed the unnecessary default input features of these models to prevent overfitting. Further details about our setup can be found in Appendix B.

For evaluation we use a rolling-window approach with a window size of 10 steps. The final evaluation metrics are the aggregated results from all 100 test windows. We report the mean squared error (MSE) and continuous ranked probability score (CRPS) (Matheson & Winkler, 1976), a commonly used score to measure how close

---

[2] https://github.com/awslabs/gluon-ts

[3] https://github.com/zalandoresearch/pytorch-ts

[4] https://github.com/ServiceNow/tactis

[5] https://github.com/thuml/Time-Series-Library

[6] https://github.com/salesforce/DeepTime

Table 2: **Quantitative Evaluation (Synthetic Data).** We compare StatiConF/DynaConF to (a) 4 univariate and (b) 6 multivariate baselines using CRPS and MSE (lower values are better). Numbers in model names indicate the number of hidden units. Full results including the standard deviations can be found in the Appendix F.

| | AR(1)-F | | AR(1)-S | | AR(1)-D | |
| Method | CRPS | MSE | CRPS | MSE | CRPS | MSE |
|---|---|---|---|---|---|---|
| GroundTruth | 0.731 | 1.2 | 0.710 | 1.6 | 0.624 | 2.1 |
| DeepAR–10 | 0.741 | 1.3 | 0.764 | 1.8 | 0.768 | 3.2 |
| DeepAR–40 | 0.740 | 1.3 | 0.776 | 1.8 | 0.820 | 3.6 |
| DeepAR–160 | 0.740 | 1.3 | 0.774 | 1.8 | 0.801 | 3.5 |
| DeepSSM | 0.755 | 1.3 | 0.761 | 1.8 | 0.803 | 3.3 |
| StatiConF–MLP | 0.753 | 1.3 | 0.784 | 1.9 | 0.764 | 3.3 |
| StatiConF–PP | 0.752 | 1.3 | 0.763 | 1.8 | 0.783 | 3.3 |
| DynaConF–MLP | 0.750 | 1.3 | 0.727 | **1.6** | 0.691 | **2.6** |
| DynaConF–PP | **0.737** | **1.2** | **0.721** | **1.6** | **0.687** | **2.6** |

[AR(1)-*: F = Flip; S = Sin; D = Dynamic]

(a) Univariate

| Method | CRPS | MSE |
|---|---|---|
| GroundTruth | 0.496 | 2.8 |
| DeepVAR–10 | 0.797 | 8.4 |
| DeepVAR–40 | 0.792 | 8.4 |
| DeepVAR–160 | 0.787 | 8.3 |
| TransformerMAF–8 | 0.800 | 8.5 |
| TransformerMAF–32 | 0.806 | 8.5 |
| TransformerMAF–128 | 0.866 | 9.4 |
| StatiConF–MLP | 0.806 | 8.5 |
| StatiConF–PP | 0.805 | 8.5 |
| DynaConF–MLP | 0.762 | 7.9 |
| DynaConF–PP | **0.609** | **4.5** |

(b) Multivariate: VAR(1)–Dynamic

the predicted distribution is to the true distribution (see Appendix E for details). Full results including standard deviations can be found in the Appendix F. In all cases lower values indicate better performance.

**Results**   The results on synthetic data are shown in Table 2. For univariate data, our full model (DynaConF) outperforms its ablated counterpart (StatiConF) consistently across datasets and encoders, validating the importance of our dynamic adaptation to non-stationary effects. DynaConF–PP is also superior to all univariate baselines, with its closest competitor DeepAR–10 behind by an average of 5.6% (CRPS). Furthermore, we note that our model with the pointwise encoder tends to outperform the MLP encoder, both for the ablated and full model. For multivariate data we observe similar trends. Here, our full model (DynaConF–PP) performs 24.3% (CRPS) better than the ablated model (StatiConF–PP) and 22.6% (CRPS) better than the best-performing baseline (DeepVAR–160). Since for synthetic data we also have access to the ground-truth models, we include the corresponding scores as references and upper bounds in terms of performance.

Figure 3 shows qualitative results of our model on AR(1)–Flip/Sin/Dynamic. Note that because the encoder in our model is non-linear, the encoding $z_t$ that is combined with $\phi_t$ is not the same as $y_{t-1}$, so the inferred $\phi_t$ may not match the sign/scale of $w_t$, although the predictions based on $\phi_t$ and $z_t$ can still stay close to $y_t = w_t y_{t-1} + \epsilon_t$, e.g., if the signs of $z_t$ and $y_{t-1}$ are flipped, and the signs of $\phi_t$ and $w_t$ are also flipped. Nonetheless, we can clearly see how $\phi_t$ differs *qualitatively* according to $w_t$: gradual changes (Figure 3b) vs. sudden jumps (Figure 3a and Figure 3c) as well as jumps to previous values (Figure 3a) vs. new values (Figure 3c). Furthermore, since our model only sees data for $t < 1000$ during training, it is remarkable that it can adapt to *unseen* changes for $t \geq 1000$ (Figure 3b and Figure 3c).

## 4.2   Experiments on Real-World Data – Set 1

**Datasets and Setup**   We evaluate the proposed method on 6 widely-used datasets[7] with published results (Lai et al., 2018; Salinas et al., 2019): (*Exchange*) daily exchange rates of 8 different countries from 1990 to 2016; (*Solar*)[8] hourly solar power production in 137 PV plants in 2006; (*Electricity*)[9] hourly electricity consumption of 370 customers from 2012 to 2014; (*Traffic*)[10] hourly occupancy data at 963 sensor locations in the San Francisco Bay area; (*Taxi*) rides taken in 30-minute intervals at 1214 locations in New York City in January 2015/2016; (*Wikipedia*) daily page views of 2000 Wikipedia articles. We use the same train/test

---

[7] https://github.com/mbohlkeschneider/gluon-ts/tree/mv_release/datasets

[8] http://www.nrel.gov/grid/solar-power-data.html

[9] https://archive.ics.uci.edu/ml/datasets/ElectricityLoadDiagrams20112014

[10] http://pems.dot.ca.gov

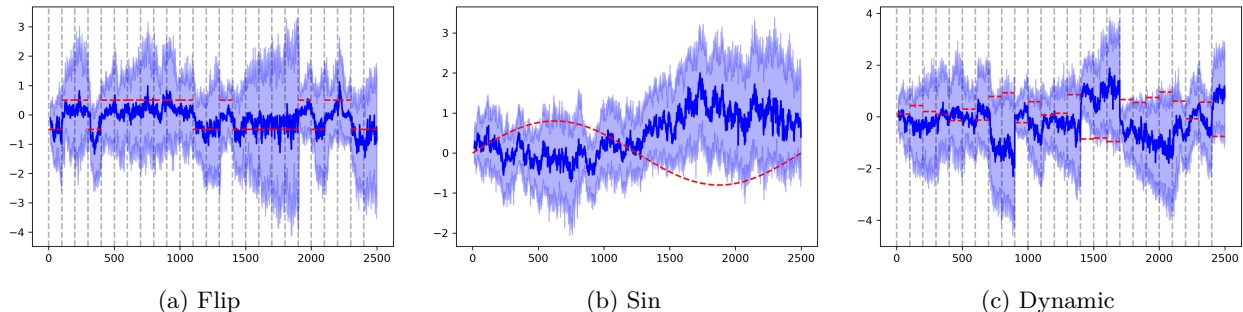

(a) Flip          (b) Sin          (c) Dynamic

Figure 3: **Qualitative Results.** We show one dimension of $\phi_t$ of DynaConF–PP inferred with particle filters at test time on (a) AR(1)–Flip, (b) AR(1)–Sin, and (c) AR(1)–Dynamic. The red dashed lines show the parameter $w_t$ in the generative processes varying over time. The blue curves and bands are the medians and 90% confidence intervals of the posterior. Note that because the encoder in our model is non-linear, the encoding $z_t$ that is combined with $\phi_t$ is not the same as $y_{t-1}$, so the inferred $\phi_t$ *may not match the sign/scale of $w_t$*, but we expect it to change gradually or suddenly according to $w_t$.

Table 3: **Quantitative Evaluation (Real-World Data – Set 1).** CRPS results of StatiConF/DynaConF and 6 probabilistic forecasting baselines on a widely used set of 6 publicly available datasets. Full results including standard deviations can be found in Appendix F. **NA**: Due to the time gap between the training and test sets for Taxi, some of the baselines cannot be applied. **OOM**: The model ran out of memory on our 16GB GPU using the minimum batch size and lookback window size.

| Method | CRPS | | | | | |
| --- | --- | --- | --- | --- | --- | --- |
| | Exchange | Solar | Electricity | Traffic | Taxi | Wikipedia |
| DeepVAR | 0.013 | 0.434 | 1.059 | 0.168 | 0.586 | 0.379 |
| GP-Copula | **0.008** | 0.371 | 0.056 | 0.133 | 0.360 | **0.236** |
| LSTM-MAF | 0.012 | 0.378 | 0.051 | 0.124 | 0.314 | 0.282 |
| TransformerMAF | 0.012 | 0.368 | 0.052 | 0.134 | 0.377 | 0.274 |
| TimeGrad | 0.009 | 0.367 | 0.049 | **0.110** | 0.311 | 0.261 |
| TACTiS | 0.011 | 0.476 | **0.047** | OOM | NA | OOM |
| StatiConF (**ours**) | 0.009 | 0.363 | 0.057 | 0.112 | **0.301** | 0.339 |
| DynaConF (**ours**) | 0.009 | **0.355** | 0.052 | 0.111 | **0.301** | 0.259 |

splits and input features, such as time of the day, as previous works with published code and results (Salinas et al., 2019; Rasul et al., 2021b;a), from which we also retrieved the performance of the baselines. For our method, we first train StatiConF and then reuse its learned encoder in DynaConF, so the optimization of DynaConF is focused on the dynamic model. Our models use a two-layer LSTM with 128 hidden units as the encoder, except for the 8-dimensional Exchange data, where the hidden size is 8. We stress again that, different from DeepVAR or LSTM-MAF, we use LSTM as an encoder of $(\boldsymbol{y}_{t-B:t-1}, \boldsymbol{x}_{t-B,t})$ only, so we actually "restart" it at every time step. More details of our hyperparameters can be found in Appendix C.

**Results** The results are shown in Table 3 and 4. We note that CRPS is only applicable to the probabilistic forecasting methods. As we can see, the relative performance of each method differs across datasets and evaluation metrics. This shows that different models may benefit from dataset-specific structure in different ways. However, DynaConF achieves the best or closest-to-the-best performance more often than the baselines. Where it does not outperform, its performance is consistently competitive. We also note that our full model (DynaConF), which adapts to changes in the conditional distribution, performs either similarly or better than our ablated model (StatiConF), which itself differs from the baselines due to its explicit modeling of time-invariant conditional distributions. Full results including standard deviations can be found in Appendix F.

Table 4: **Quantitative Evaluation (Real-World Data – Set 1).** MSE results of StatiConF/DynaConF and 9 baselines on a widely used set of 6 publicly available datasets. Full results including standard deviations can be found in Appendix F. **NA**: Due to the time gap between the training and test sets for Taxi, some of the baselines cannot be applied. **OOM**: The model ran out of memory on our 16GB GPU using the minimum batch size and lookback window size.

| Method | MSE | | | | | |
| --- | --- | --- | --- | --- | --- | --- |
| | Exchange [e-4] | Solar [e+2] | Electricity [e+5] | Traffic [e-4] | Taxi [e+1] | Wikipedia [e+7] |
| DeepVAR | 1.6 | 9.3 | 2.1 | 6.3 | 7.3 | 7.2 |
| GP-Copula | 1.7 | 9.8 | 2.4 | 6.9 | 3.1 | 4.0 |
| LSTM-MAF | 3.8 | 9.8 | 1.8 | 4.9 | 2.4 | 3.8 |
| TransformerMAF | 3.4 | 9.3 | 2.0 | 5.0 | 4.5 | **3.1** |
| TimeGrad | 2.5 | 8.8 | 2.0 | **4.2** | 2.6 | 3.8 |
| TACTiS | 2.6 | 14 | **1.4** | OOM | NA | OOM |
| NS Transformer | 2.5 | 10 | 2.2 | 7.0 | NA | 5.2 |
| DeepTime | **1.4** | 9.6 | 2.4 | 4.8 | NA | 4.6 |
| Koopa | **1.4** | 10 | 2.4 | 5.8 | NA | 3.9 |
| StatiConF (**ours**) | 2.3 | 8.2 | 1.8 | 4.8 | **2.2** | 4.0 |
| DynaConF (**ours**) | 2.0 | **8.0** | 1.7 | 4.8 | **2.2** | 3.7 |

Table 5: **Quantitative Evaluation (Real-World Data – Set 2).** CRPS results of StatiConF/DynaConF and 6 probabilistic forecasting baselines on publicly available sales and temperature data.

| Method | CRPS | |
| --- | --- | --- |
| | Walmart | Temperature |
| DeepVAR | $0.635_{\pm 0.014}$ | $0.584_{\pm 0.023}$ |
| GP-Copula | $0.557_{\pm 0.011}$ | $0.209_{\pm 0.006}$ |
| LSTM-MAF | $0.685_{\pm 0.019}$ | $0.224_{\pm 0.006}$ |
| TransformerMAF | $0.638_{\pm 0.060}$ | $0.244_{\pm 0.017}$ |
| TimeGrad | $0.604_{\pm 0.026}$ | $0.195_{\pm 0.002}$ |
| TACTiS | $0.618_{\pm 0.047}$ | $\mathbf{0.183}_{\pm 0.002}$ |
| StatiConF (**ours**) | $0.573_{\pm 0.015}$ | $0.199_{\pm 0.004}$ |
| DynaConF (**ours**) | $\mathbf{0.512}_{\pm 0.001}$ | $0.186_{\pm 0.003}$ |

## 4.3 Experiments on Real-World Data – Set 2

**Datasets and Setup**  We further evaluate our method against state-of-the-art baselines on two more publicly available datasets: (*Walmart*)[11] weekly sales of 45 Walmart stores from February 2010 to October 2012; (*Temperature*)[12] monthly average temperatures of 1000 cities from January 1980 to September 2020. We use the last 10% of the training time periods as validation sets to tune the hyperparameters for all models. For additional details about the experiment setup we refer to Appendix D.

**Results**  The results are shown in Table 5 and 6. On these datasets, DynaConF shows a clear advantage over the baselines in terms of both CRPS and MSE (except TACTiS on Temperature). We believe this is mainly due to more significant changes in the conditional distributions of the time series in Set 2, a hypothesis which is also supported by the superior performance of DynaConF compared to the ablated StatiConF model. It is also worth noting that the best-performing baseline models across Set 1 and Set 2 are different, while the proposed model performs more consistently. Combined with the results in the previous section, we conclude that DynaConF performs competitively if the conditional distribution remains relatively stable but outperforms the baselines if the conditional distribution does undergo dynamic changes, in line with its design and ability to account for such changes.

---

[11]https://www.kaggle.com/datasets/yasserh/walmart-dataset
[12]https://www.kaggle.com/datasets/hansukyang/temperature-history-of-1000-cities-1980-to-2020

Table 6: **Quantitative Evaluation (Real-World Data – Set 2).** MSE results of StatiConF/DynaConF and 9 baselines on publicly available sales and temperature data.

| Method | MSE | |
|---|---|---|
| | Walmart [e-1] | Temperature [e-1] |
| DeepVAR | $2.921_{\pm 4.846e\text{-}3}$ | $5.716_{\pm 4.517e\text{-}2}$ |
| GP-Copula | $2.761_{\pm 1.505e\text{-}2}$ | $1.432_{\pm 1.088e\text{-}2}$ |
| LSTM-MAF | $4.275_{\pm 2.709e\text{-}2}$ | $1.630_{\pm 9.890e\text{-}3}$ |
| TransformerMAF | $3.651_{\pm 8.102e\text{-}2}$ | $1.950_{\pm 2.828e\text{-}2}$ |
| TimeGrad | $3.190_{\pm 2.131e\text{-}2}$ | $1.077_{\pm 2.341e\text{-}3}$ |
| TACTiS | $5.666_{\pm 4.668e\text{-}1}$ | $\mathbf{0.952}_{\pm 1.738e\text{-}3}$ |
| NS Transformer | $2.603_{\pm 3.735e\text{-}3}$ | $1.719_{\pm 4.329e\text{-}3}$ |
| DeepTime | $3.798_{\pm 6.788e\text{-}2}$ | $1.664_{\pm 1.517e\text{-}2}$ |
| Koopa | $2.761_{\pm 3.206e\text{-}3}$ | $1.170_{\pm 3.523e\text{-}3}$ |
| StatiConF (**ours**) | $3.146_{\pm 1.900e\text{-}2}$ | $1.241_{\pm 8.489e\text{-}3}$ |
| DynaConF (**ours**) | $\mathbf{2.497}_{\pm 6.840e\text{-}3}$ | $1.017_{\pm 2.681e\text{-}3}$ |

## 5 Limitations

Our model currently has the following limitations: (1) since the variational posterior model complexity scales in $O(T)$, it could be challenging to train the model on extremely long time series; and (2) we used a simple observation distribution family in this work to allow efficient inference, but there are cases where this may impact performance. For future work, it would be interesting to develop new variational posteriors and training algorithms that can scale better and more flexible inference algorithms that can deal with more complex observation distributions.

## 6 Conclusion

In this work, we addressed the problem of modeling and forecasting time series with non-stationary conditional distributions. We proposed a new model, DynaConF, that explicitly decouples the time-invariant conditional distribution modeling and the time-variant non-stationarity modeling. We designed specific architectures, developed new types of variational posteriors, and employed Rao-Blackwellized particle filters to allow the model to train efficiently on large multivariate time series and adapt to unseen changes at test time. Results on synthetic and real-world data show that our model can learn and adapt to different types of changes (continuous or discontinuous) in the conditional distribution and performs competitively or better than state-of-the-art time series forecasting models.

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

## A    Optimization Procedure

In contrast to existing time series models, we utilize a flexible variational posterior with a large number of parameters in the order of $O(T)$ and a structured prior model to account for conditional distribution changes over time. However, jointly optimizing over these variational parameters and the parameters in the conditional distribution model itself using stochastic gradient descent can be prohibitively demanding on the computational resources, especially GPU memory. Instead, we propose an alternative optimization procedure to learn these parameters.

Specifically, instead of optimizing by stochastic gradient descent (SGD) over all parameters in both the conditional distribution model and the prior and variational posterior model, we propose to learn the model by alternating the optimization of the parameters in the conditional distribution model and the prior and variational posterior models. For the former, we condition on the samples from the current posterior model while optimizing the conditional distribution model on randomly sampled sub-sequences from the time series using SGD. For the latter, we fix the conditional distribution model, sample from the variational posterior model over the entire time series either sequentially or in parallel, depending on the variational posterior model we use, and compute the loss using the samples through the conditional distribution, prior, and posterior models. Then we perform an update on the prior and posterior model parameters using the gradient from the loss.

When the time series is high-dimensional and GPU memory becomes a constraint during training, we randomly sample a subset of *observation* dimensions for each batch, since the loss decomposes over the observation dimensions in our model.

For our models, we use Adam (Kingma & Ba, 2014) as the optimizer with the default initial learning rate of 0.001 unless it is chosen using the validation set. The dimension of the latent vector $z_{t,i}$ (see Section 3.2) is set to $E = 4$ across all the experiments.

## B    Synthetic Data: Details and Hyperparameters

On synthetic datasets, we use the validation set to early stop and choose the best model for both the baselines and our models. We perform 50 updates per epoch. We use 32 hidden units for our 2-layer MLP encoder. For the baselines, we report their results with different hidden sizes, including the default ones.

We keep most baseline hyperparameters to their default values but make the following changes to account for properties of our synthetic data: (1) To reduce overfitting, we remove any unnecessary input features from the models and use only past observations with time lag 1 as input. (2) To allow the models to adapt to changes in the conditional distribution we increase the lookback window size to 200. This allows the models to see enough observations generated with the latest ground-truth distribution parameters, so the models have the necessary information to adapt to the current distribution. For VAR(1)–Dynamic, we also tried extending it to 500, but it did not improve the performance. (3) DeepSSM allows modeling of trend and seasonality, but since our synthetic data do not have those, we explicitly remove those components from the model specification to avoid overfitting; (4) DeepVAR allows modeling of different covariance structures in the noise, such as diagonal, low-rank, and full-rank. Since our synthetic data follow a diagonal covariance structure in the noise, we explicitly specify it for DeepVAR.

## C    Real-World Data – Set 1: Details and Hyperparameters

On real-world datasets with published results, we use the last 10% of the training time period as the validation set and choose the initial learning rate, number of training epochs, and model sizes using the performance on the validation set for StatiConF. After training StatiConF, we reuse its encoders in DynaConF, so it only needs to learn the dynamic model. For DynaConF, we use the same validation set to choose the number of training epochs and use 0.01 as the initial learning rate.

Because of the diversity of the real-world datasets, we further apply techniques to stablize training. Specifically, for all datasets, we use the mean and standard deviation to shift and scale each dimension of the time series.

For Exchange, we use the mean and standard deviation of the recent past data in a moving lookback window. For the other datasets, we simply use the global mean and standard deviation of each dimension computed using the whole training set. In all cases, for forecasting, the output from the model is inversely scaled and shifted back for evaluation. These design choices were made based on the performance on the validation sets.

On real-world datasets, extreme values or outliers may cause instability during training. We optionally apply Winsorization using the quantiles (0.025 and 0.975) computed from the recent past data in a moving lookback window on Traffic and Wikipedia. The decisions of whether to apply this transformation were based on the results on the validation sets.

For TACTiS, NS Transformer, DeepTime, and Koopa, we use the same hyperparameters as in their recommended settings for similar datasets from their published results. For the other baselines, the results are retrieved from previous publications (Rasul et al., 2021a;b).

## D  Real-World Data – Set 2: Details and Hyperparameters

On the Walmart and Temperature datasets, the experiment setup is the same for all the models. For Walmart, the forecast window size is 4 weeks, and the test set consists of the last 20 weeks. For Temperature, the forecast window size is 3 months, and the test set consists of the last 24 months. We found it helpful to normalize the time series using the means and standard deviations computed from the training set for each dataset for stabilizing training, especially for LSTM-MAF and TransformerMAF. We use the last 10% of the training time period as the validation set to tune the hyperparameters, including the model size $(32, 128, 512, 2048)$ and initial learning rate $(0.01, 0.001)$, and for early stopping for our models and the first 5 baseline models, which share similar encoder architectures as ours. For the baseline models, we also use it to choose whether to apply marginal transformation (Salinas et al., 2019) and mean scaling (Salinas et al., 2020). Out of all the baseline models, we found NS Transformer, DeepTime and Koopa to be sensitive to the lookback window size, which we tuned across a regular grid of multipliers of the forecast window size. For the other baselines, we found that in many cases increasing the window size resulted in worse validation performance compared to using the default size. In the other cases, it resulted in small improvements on the validation set but similar or worse performance on the test set. For consistency, we settled on using the same default lookback window size.

## E  Details on Evaluation

We run all experiments for three different random seeds independently and calculate and report the mean and standard deviation of each evaluation metric for each model. On synthetic datasets, we use 1000 sample paths to empirically estimate the predicted distributions for all models. On real-world datasets, we use 100 sample paths.

We use two evaluation metrics: mean squared error (MSE) and continuous ranked probability score (CRPS)(Matheson & Winkler, 1976). Assume that we observe $y$ at time $t$ but a probabilistic forecasting model predicts the distribution of $y$ to be $F$. MSE is widely used for time series forecasting, and for a probabilistic forecasting model, where the mean of the distribution is used for point prediction, it is defined as

$$\mathrm{MSE}(F, y) = (\mathrm{E}_{z \sim F}[z] - y)^2 \tag{16}$$

for a single time point $t$ and averaged over all the time points in the test set.

CRPS has been used for evaluating how close the predicted probability distribution is to the ground-truth distribution and is defined as

$$\mathrm{CRPS}(F, y) = \int_{\mathbb{R}} (F(z) - \mathbb{I}[y \leq z])^2 \mathrm{d}z, \tag{17}$$

for a single time point $t$ and averaged over all the time points in the test set, where $\mathbb{I}$ denotes the indicator function. Generally, $F(z)$ can be approximated by the empirical distribution of the samples from the predicted distribution. Both MSE and CRPS can be applied to multivariate time series by computing the metric on each dimension and then averaging over all the dimensions.

Table 7: **Quantitative Evaluation (Synthetic Data).** CRPS results on univariate processes AR(1)-Flip/Sin/Dynamic.

| Method | CRPS | | |
| --- | --- | --- | --- |
| | AR(1)-F | AR(1)-S | AR(1)-D |
| GroundTruth | $0.731_{\pm 0.001}$ | $0.710_{\pm 0.001}$ | $0.624_{\pm 0.001}$ |
| DeepAR–10 | $0.741_{\pm 0.005}$ | $0.764_{\pm 0.004}$ | $0.768_{\pm 0.011}$ |
| DeepAR–40 | $0.740_{\pm 0.003}$ | $0.776_{\pm 0.002}$ | $0.820_{\pm 0.053}$ |
| DeepAR–160 | $0.740_{\pm 0.001}$ | $0.774_{\pm 0.004}$ | $0.801_{\pm 0.047}$ |
| DeepSSM | $0.755_{\pm 0.001}$ | $0.761_{\pm 0.001}$ | $0.803_{\pm 0.001}$ |
| StatiConF–MLP | $0.753_{\pm 0.001}$ | $0.784_{\pm 0.002}$ | $0.764_{\pm 0.003}$ |
| StatiConF–PP | $0.752_{\pm 0.002}$ | $0.763_{\pm 0.001}$ | $0.783_{\pm 0.002}$ |
| DynaConF–MLP | $0.750_{\pm 0.001}$ | $0.727_{\pm 0.019}$ | $0.691_{\pm 0.006}$ |
| DynaConF–PP | $\mathbf{0.737}_{\pm 0.001}$ | $\mathbf{0.721}_{\pm 0.005}$ | $\mathbf{0.687}_{\pm 0.010}$ |

[AR(1)-*: F = Flip; S = Sin; D = Dynamic]

Table 8: **Quantitative Evaluation (Synthetic Data).** MSE results on univariate processes AR(1)-Flip/Sin/Dynamic.

| Method | MSE | | |
| --- | --- | --- | --- |
| | AR(1)-F | AR(1)-S | AR(1)-D |
| GroundTruth | $1.2_{\pm 9.2e\text{-}4}$ | $1.6_{\pm 3.0e\text{-}3}$ | $2.1_{\pm 5.3e\text{-}3}$ |
| DeepAR–10 | $1.3_{\pm 1.7e\text{-}2}$ | $1.8_{\pm 1.8e\text{-}2}$ | $3.2_{\pm 7.2e\text{-}2}$ |
| DeepAR–40 | $1.3_{\pm 8.9e\text{-}3}$ | $1.8_{\pm 1.3e\text{-}2}$ | $3.6_{\pm 3.8e\text{-}1}$ |
| DeepAR–160 | $1.3_{\pm 2.8e\text{-}3}$ | $1.8_{\pm 9.6e\text{-}3}$ | $3.5_{\pm 3.3e\text{-}1}$ |
| DeepSSM | $1.3_{\pm 2.2e\text{-}3}$ | $1.8_{\pm 2.8e\text{-}3}$ | $3.3_{\pm 3.0e\text{-}3}$ |
| StatiConF–MLP | $1.3_{\pm 4.0e\text{-}3}$ | $1.9_{\pm 7.7e\text{-}3}$ | $3.3_{\pm 3.5e\text{-}2}$ |
| StatiConF–PP | $1.3_{\pm 4.7e\text{-}3}$ | $1.8_{\pm 5.4e\text{-}3}$ | $3.3_{\pm 6.2e\text{-}3}$ |
| DynaConF–MLP | $1.3_{\pm 4.0e\text{-}3}$ | $\mathbf{1.6}_{\pm 9.2e\text{-}2}$ | $\mathbf{2.6}_{\pm 5.2e\text{-}2}$ |
| DynaConF–PP | $\mathbf{1.2}_{\pm 5.5e\text{-}3}$ | $\mathbf{1.6}_{\pm 2.7e\text{-}2}$ | $\mathbf{2.6}_{\pm 5.1e\text{-}2}$ |

[AR(1)-*: F = Flip; S = Sin; D = Dynamic]

## F   Additional Experiment Results

Table 7, 8, 9, and 10 show the full CRPS and MSE results of the baselines and our models on the univariate and multivariate processes respectively. Table 11 and 12 show the full CRPS and MSE results of our models with means and standard deviations on the real-world datasets. The results of the first 5 baselines on the real-world data – set 1 are from (Rasul et al., 2021a;b).

## G   Qualitative Comparison of DynaConF and StatiConF

Here we show some qualitative comparison of DynaConF and StatiConF predictions to highlight the differences in their forecasting behaviors when facing changes in the distribution. Specifically, we compare their predictions on one dimension from the Electricity dataset, where there appears to be a structural change in the time series at test time. As we can see in Figure 4, as the forecast window keeps moving forward, DynaConF quickly adapts to the new distribution, while StatiConF struggles to react. We note that the encoder is shared between these two models, so the differences come from the non-stationary model part.

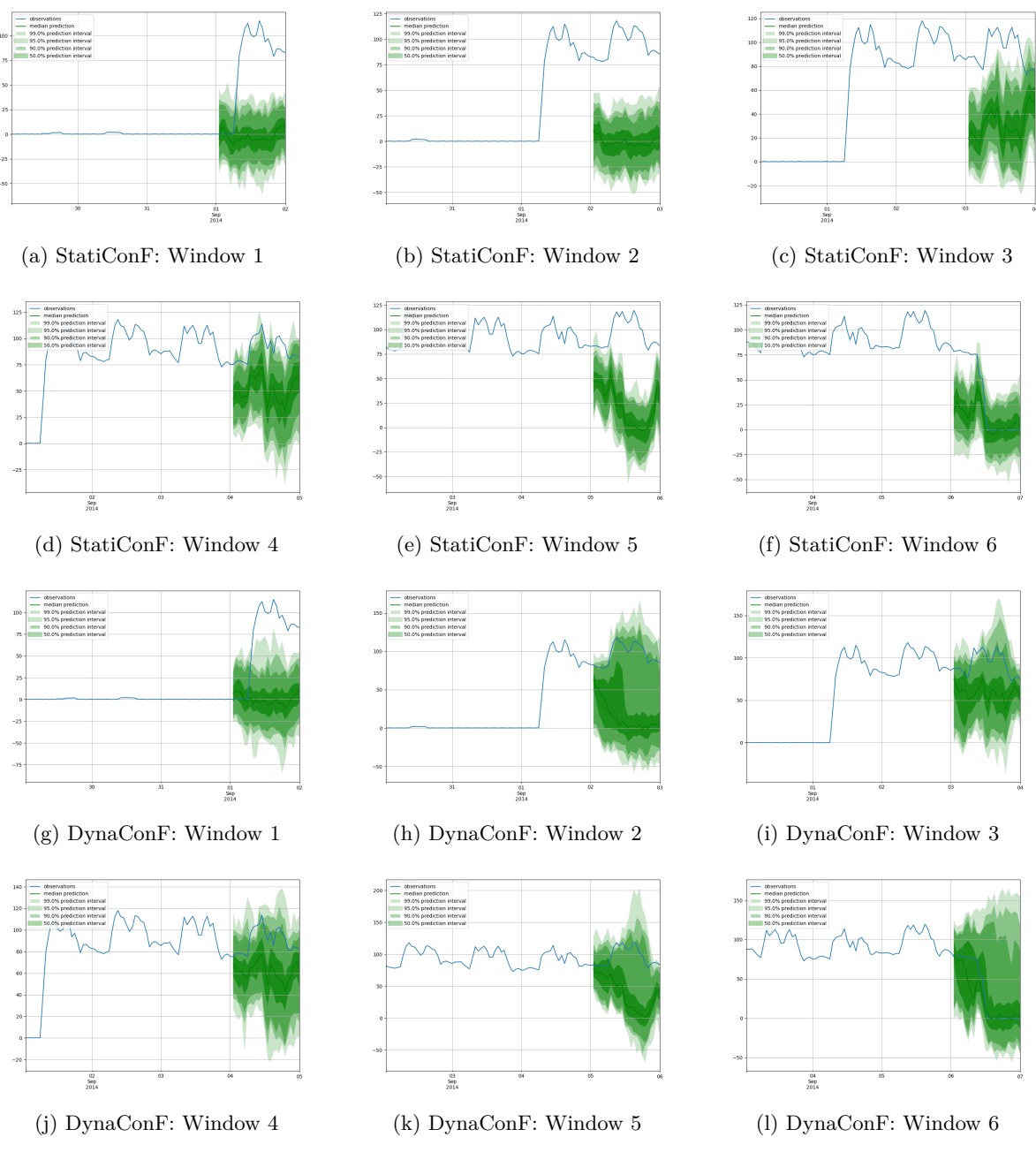

Figure 4: **Qualitative Results on Electricity Dataset.** We show the predictions of StatiConF (a–f) vs DynaConF (g–l) on one dimension of electricity dataset, where there appears to be a change in the distribution at test time.

Table 9: **Quantitative Evaluation (Synthetic Data).** CRPS results on multivariate process VAR(1)-Dynamic.

| Method | CRPS |
|---|---|
| GroundTruth | $0.496_{\pm 0.001}$ |
| DeepVAR–10 | $0.797_{\pm 0.009}$ |
| DeepVAR–40 | $0.792_{\pm 0.000}$ |
| DeepVAR–160 | $0.787_{\pm 0.001}$ |
| TransformerMAF–8 | $0.800_{\pm 0.001}$ |
| TransformerMAF–32 | $0.806_{\pm 0.008}$ |
| TransformerMAF–128 | $0.866_{\pm 0.077}$ |
| StatiConF–MLP | $0.806_{\pm 0.004}$ |
| StatiConF–PP | $0.805_{\pm 0.002}$ |
| DynaConF–MLP | $0.762_{\pm 0.036}$ |
| DynaConF–PP | $\mathbf{0.609}_{\pm 0.012}$ |

(StatiConF–MLP through DynaConF–PP marked as **(ours)**)

Table 10: **Quantitative Evaluation (Synthetic Data).** MSE results on multivariate process VAR(1)-Dynamic.

| Method | MSE |
|---|---|
| GroundTruth | $2.8_{\pm 7.6e\text{-}3}$ |
| DeepVAR–10 | $8.4_{\pm 5.6e\text{-}2}$ |
| DeepVAR–40 | $8.4_{\pm 2.7e\text{-}3}$ |
| DeepVAR–160 | $8.3_{\pm 2.2e\text{-}2}$ |
| TransformerMAF–8 | $8.5_{\pm 2.9e\text{-}2}$ |
| TransformerMAF–32 | $8.5_{\pm 3.7e\text{-}2}$ |
| TransformerMAF–128 | $9.4_{\pm 1.2e0}$ |
| StatiConF–MLP | $8.5_{\pm 8.5e\text{-}2}$ |
| StatiConF–PP | $8.5_{\pm 2.3e\text{-}2}$ |
| DynaConF–MLP | $7.9_{\pm 5.7e\text{-}1}$ |
| DynaConF–PP | $\mathbf{4.5}_{\pm 3.0e\text{-}1}$ |

(StatiConF–MLP through DynaConF–PP marked as **(ours)**)

Table 11: **Quantitative Evaluation (Real-World Data – Set 1).** CRPS results with means and standard deviations on 6 publicly available datasets.

| Method | CRPS | | | | | |
|---|---|---|---|---|---|---|
| | Exchange | Solar | Electricity | Traffic | Taxi | Wikipedia |
| DeepVAR | $0.013_{\pm 0.000}$ | $0.434_{\pm 0.012}$ | $1.059_{\pm 0.001}$ | $0.168_{\pm 0.037}$ | $0.586_{\pm 0.004}$ | $0.379_{\pm 0.004}$ |
| GP-Copula | $\mathbf{0.008}_{\pm 0.000}$ | $0.371_{\pm 0.022}$ | $0.056_{\pm 0.002}$ | $0.133_{\pm 0.001}$ | $0.360_{\pm 0.201}$ | $\mathbf{0.236}_{\pm 0.000}$ |
| LSTM-MAF | $0.012_{\pm 0.003}$ | $0.378_{\pm 0.032}$ | $0.051_{\pm 0.000}$ | $0.124_{\pm 0.002}$ | $0.314_{\pm 0.003}$ | $0.282_{\pm 0.002}$ |
| TransformerMAF | $0.012_{\pm 0.003}$ | $0.368_{\pm 0.001}$ | $0.052_{\pm 0.000}$ | $0.134_{\pm 0.001}$ | $0.377_{\pm 0.002}$ | $0.274_{\pm 0.007}$ |
| TimeGrad | $0.009_{\pm 0.001}$ | $0.367_{\pm 0.001}$ | $0.049_{\pm 0.002}$ | $\mathbf{0.110}_{\pm 0.002}$ | $0.311_{\pm 0.030}$ | $0.261_{\pm 0.020}$ |
| TACTiS | $0.011_{\pm 0.000}$ | $0.476_{\pm 0.005}$ | $\mathbf{0.047}_{\pm 0.000}$ | OOM | NA | OOM |
| StatiConF | $0.009_{\pm 0.001}$ | $0.363_{\pm 0.019}$ | $0.057_{\pm 0.001}$ | $0.112_{\pm 0.007}$ | $\mathbf{0.301}_{\pm 0.008}$ | $0.339_{\pm 0.011}$ |
| DynaConF | $0.009_{\pm 0.000}$ | $\mathbf{0.355}_{\pm 0.008}$ | $0.052_{\pm 0.002}$ | $0.111_{\pm 0.007}$ | $\mathbf{0.301}_{\pm 0.007}$ | $0.259_{\pm 0.001}$ |

Table 12: **Quantitative Evaluation (Real-World Data − Set 1).** MSE results with means and standard deviations on 6 publicly available datasets.

| Method | MSE | | | | | |
| --- | --- | --- | --- | --- | --- | --- |
| | Exchange [e-4] | Solar [e+2] | Electricity [e+5] | Traffic [e-4] | Taxi [e+1] | Wikipedia [e+7] |
| DeepVAR | 1.6 | 9.3 | 2.1 | 6.3 | 7.3 | 7.2 |
| GP-Copula | 1.7 | 9.8 | 2.4 | 6.9 | 3.1 | 4.0 |
| LSTM-MAF | 3.8 | 9.8 | 1.8 | 4.9 | 2.4 | 3.8 |
| TransformerMAF | 3.4 | 9.3 | 2.0 | 5.0 | 4.5 | **3.1** |
| TimeGrad | 2.5 | 8.8 | 2.0 | **4.2** | 2.6 | 3.8 |
| TACTiS | $2.6_{\pm 1.1e\text{-}5}$ | $14_{\pm 9.3e+1}$ | $\mathbf{1.4}_{\pm 3.4e+3}$ | OOM | NA | OOM |
| NS Transformer | $2.5_{\pm 4.6e\text{-}5}$ | $10_{\pm 1.4e+2}$ | $2.2_{\pm 6.8e+3}$ | $7.0_{\pm 9.3e\text{-}6}$ | NA | $5.2_{\pm 1.1e+7}$ |
| DeepTime | $\mathbf{1.4}_{\pm 7.7e\text{-}7}$ | $9.6_{\pm 7.9e+0}$ | $2.4_{\pm 1.1e+4}$ | $4.8_{\pm 2.1e\text{-}6}$ | NA | $4.6_{\pm 9.7e+5}$ |
| Koopa | $\mathbf{1.4}_{\pm 7.8e\text{-}6}$ | $10_{\pm 5.2e+1}$ | $2.4_{\pm 1.6e+4}$ | $5.8_{\pm 2.2e\text{-}5}$ | NA | $3.9_{\pm 5.7e+5}$ |
| StatiConF | $2.3_{\pm 4.7e\text{-}5}$ | $8.2_{\pm 8.8e+1}$ | $1.8_{\pm 2.4e+4}$ | $4.8_{\pm 6.2e\text{-}6}$ | $\mathbf{2.2}_{\pm 9.9e\text{-}1}$ | $4.0_{\pm 4.0e+5}$ |
| DynaConF | $2.0_{\pm 1.175e\text{-}5}$ | $\mathbf{8.0}_{\pm 3.7e+1}$ | $1.7_{\pm 3.0e+4}$ | $4.8_{\pm 7.3e\text{-}6}$ | $\mathbf{2.2}_{\pm 9.2e\text{-}1}$ | $3.7_{\pm 4.1e+5}$ |

