# OpenReview forum: "DynaConF: Dynamic Forecasting of Non-Stationary Time Series"
_TMLR — Accepted by TMLR_

### Review · Reviewer_PyiA · 2023-11-20

**Summary Of Contributions:**

This paper proposes a novel approach to address the prediction of non-stationary time series by combining the time-variant (non-stationary) and time-invariant components (stationary) within the time series. The time-invariant part is modeled with fixed parameters $\psi$, while the time-variant part represents non-stationary information, controlled by time-varying parameters $\phi_t$.

**Audience:**

Yes

**Broader Impact Concerns:**

Not applicable.

**Claims And Evidence:**

No

**Requested Changes:**

See the weaknesses above.

**Strengths And Weaknesses:**

## Strengths

This paper introduces a novel method for time series prediction, incorporating both time-invariant and time-varying information within the model.

## Weaknesses
1. This paper lacks a reasonable explanation of the parameterization formulas for non-smooth models. For example, in equation (3), it is not clear why $z_{t,i}$ is transformed into the distribution parameters of $y_t$ with parameters related to time information. Furthermore, in equation (5), there is no clear justification as to why a Gaussian distribution is chosen as the initial distribution of $\mathcal{X}_t$ and the distribution of $\epsilon_t$.
2. There is a lack of theoretical analysis of the identifiability of the proposed model.
3. What is the novelty of the models and methods proposed in this paper? I am concerned that the contribution of this paper may not be sufficient.

---

> ### Author Response · Authors · 2023-12-12
>
> ## Lack of Justifications
>
> The design choices in our models such as parametrization and distribution assumptions are driven by simplicity and efficiency. We tried to choose the simplest design that achieves our goal of separating stationary and non-stationary modeling in a deep time series model. That is why we chose a linear transformation in Eq (3) to map the latent variable $\boldsymbol{z}_{t,i}$ to the mean and variance of the observations. Furthermore, we chose the Gaussian distribution as initial distribution of $\boldsymbol{\chi}_t$ and $\boldsymbol{\epsilon}_t$, because it enables efficient inference via Rao-Blackwellization as mentioned in Section 3.5.
>
> ## Identifiability of the Model
>
> Identifiability of *deep* time series models is a very interesting question, however, not one typically addressed in papers in this area. The scope of such a paper would be different and deserves an analysis in a separate work.
>
> ## Novelty
>
> The novelty in our work lies in the decoupling of the stationary and non-stationary part in a *deep* time series model. As described in the introduction, existing deep time series models either assume a stationary *conditional* distribution for forecasting or utilize a recurrent structure, e.g., an RNN, and assume that it is capable of accounting for non-stationarity implicitly. In our work, we explicitly decouple the stationary and non-stationary components at the architectural level, allowing us to have strong inductive biases in both, which leads to improved modeling of non-stationary conditional distributions for time series.

---

### Review · Reviewer_Gk9X · 2023-11-30

**Summary Of Contributions:**

This paper tackles the modelling the modelling of non-stationary conditional distributions in time series by decoupling the modelling of time-invariant and time-variant parts of the conditional distribution. The time-invariant part models a stationary conditional distribution, given a control variable, while the time-variant part focuses on modeling the changes in this conditional distribution over time through the control variable. The experimental results on synthetic and real-world datasets show that the model can adapt well to non-stationary time series.

**Audience:**

Yes

**Broader Impact Concerns:**

No broader impact concerns.

**Claims And Evidence:**

No

**Requested Changes:**

The paper needs several revisions to be ready for acceptance. Addressing the weaknesses is important for me to recommend the acceptance of this paper.

**Strengths And Weaknesses:**

## Strengths

* **Paper is structured well**: The idea is well-presented, and the paper is well-written. Especially in Section 3, the writing flows very well.
* **Related work is very elaborate**: The authors discuss non-stationarity in the marginal distributions and conditional distributions well in the related work. I also like how the authors compare the problem to the online learning setup and make the distinction.
* **Evaluation on a comprehensive set of datasets**: The authors evaluate their method on a comprehensive set of datasets comprising toy datasets and real-world datasets.

## Weaknesses

* **Not multivariate**: The paper claims to propose a multivariate time series modelling method. However, this is not correct as evident from

1. The model: The model does not model correlations between series anywhere.

> We construct the distribution of $y_t$ such that each dimension $i$ of $y_t$, denoted as $y_{t, i}$, is conditionally independent of the others given $h_t$;

> We propose to separate $\chi_t$ along the dimensions of $y_t$ into $D_{y}$ groups. For each $i = 1, . . . , D_{y}$, we define $\chi_{t,i} \in R^{E}$ as in Eq. 5. The final $\chi_t$ is the concatenation of $\chi_{t,i}$ for all $i$. The intuition is to allow the group of components of $\chi_t$ modulating each dimension of $y_t$ to change independently of the others, corresponding to the conditional independence assumption we made in Section 3.2.

The method assumes that the different series are independent. This corresponds to the model being a univariate model. Even if it has the ability to consume multivariate time series in its input, this conditional independence assumption makes it a univariate model.

 2. The evaluation

The model evaluates on the CRPS metric for probabilistic forecasting. However, this metric is a univariate metric that only measures the quality of the marginal distributions of each series learned.

The CRPS-Sum metric is the one used for evaluating multivariate models [5,8,9]. Further, Energy Score is also used [10]. If the method claims to be a multivariate method, the model has to be evaluated and compared to baselines with these metrics.

3. No qualitative evidence of correlations being learned

If I am still missing something and the model is indeed a multivariate method, the authors should compute the correlations between the different series predicted and plot the correlations learned for various datasets. Only if these correlations are meaningful, it would mean that the method in fact models correlations between the various setup

* **Missing recent important baselines**:
1. The paper misses comparing to baselines that specifically target non-stationary time series [1, 2, 3, 4]. I am aware [3] does not have published code as per their paper. But the authors must compare to [1,2,4] as they have open-source code available.
2. The paper misses many recent baselines on probabilistic forecasting. The latest baseline that the paper uses is one published in 2021 [8]. The paper should compare to at least 2 baselines from [5, 6, 7] to claim that it is state-of-the-art.


* **Lack of qualitative evaluation**: There is no qualitative evaluation of the model in the paper, such as example forecast predictions of the model. Specifically since this paper tackles non-stationary time series, it is important to point out cases in real-world time series where other methods fail to model non-stationarity.

### Minor issues

* **Confusing Notation**:

> $x_t ∈ R^{D_x}$ is an input containing contextual information

From this statement, my interpretation is that $x_{1:T +H}$ contains the time series history given as context.

> At test time, we are given past observations $y_{1:T}$ as well as input features $x_{1:T +H}$, including $H$ future steps, to infer the conditional distribution $p(y_{T +1:T +H}|y_{1:T} , x_{1:T +H})$. We note again that $x_{T +1:T +H}$ only contains information known ahead, such as day of the week.

However, from this statement, it means that $x_{T +1:T +H}$ contains the covariates too?

Please also see below the related question about the confusion about the setup of forecasting considered.

* **Issues with several sentences in the paper**:
> Despite this exciting progress, current time series forecasting methods often make the implicit assumption that the distribution of the observed time series conditioned on the input and past is time-invariant.

Not true; not all papers make this assumption. For instance, at least the papers listed in the “Missing citations” point explicitly address this assumption.

> When these models are applied to, and therefore conditioned on, the entire history of the time series, they can theoretically “memorize” different conditional distributions at different points in time

This is a problematic statement since memorization can mean different things. The authors should rephrase this.

* **Confusing setup of forecasting**:
> We study the problem of modeling and forecasting time series with changes in the conditional distribution $p(y_t|y_{t−B:t−1}, x_{t−B:t})$ over time $t$, where $y_t \in R^{D_y}$ is a target time series, and $x_t \in R^{D_x}$ is an input containing contextual information.

As per the above statement, it seems that the method takes as input some time series of the data, and predicts other target time series. However, typical multivariate time series papers [5,6,7] predict the same series for future time horizons ($R^{D_x} = R^{D_y}$).

Can the authors clarify if this is the case, or rephrase the statement and notations? If this is the case, can the authors explain why they chose this setup?

* **Missing citations**:

The paper misses some key citations on papers tackling non-stationarity in time series [1,2,3,4]. The authors should discuss and contrast their method with these methods.
The paper also misses key papers in multivariate probabilistic forecasting [5, 6, 7].

## Questions

* **Unrolling of the RNN**:  We note that a key distinction between our model’s use of an RNN and a typical deep time series model using an RNN is that the latter keeps unrolling the RNN over time to model the dynamics of the time series. In contrast, we unroll the RNN for $B + 1$ steps to summarize $(y_{t−B:t−1}, x_{t−B:t})$ in the exact same way at each time point t, i.e., we apply it in a time-invariant manner.

How is it possible to unroll an RNN the exact same way at each time point? I do not understand how the RNN is applied in a time-invariant manner here.

## References

[1] Chen, Xinyu, et al. "Nonstationary temporal matrix factorization for multivariate time series forecasting." arXiv preprint arXiv:2203.10651 (2022).

[2] Liu, Yong et al. “Non-stationary Transformers: Exploring the Stationarity in Time Series Forecasting.” Neural Information Processing Systems (2022).

[3] Yanchenko, Anna K., and Sayan Mukherjee. "Stanza: a nonlinear state space model for probabilistic inference in non-stationary time series." arXiv preprint arXiv:2006.06553 (2020).

[4] Woo, Gerald, et al. "Deeptime: Deep time-index meta-learning for non-stationary time-series forecasting." arXiv preprint arXiv:2207.06046 (2022).

[5] Drouin, Alexandre, Étienne Marcotte, and Nicolas Chapados. "Tactis: Transformer-attentional copulas for time series." International Conference on Machine Learning. PMLR, 2022.

[6] Lopez Alcaraz, Juan Miguel, and Nils Strodthoff. "Diffusion-based time series imputation and forecasting with structured atate apace models." Transactions on machine learning research (2023): 1-36.

[7] Tashiro, Yusuke, et al. "Csdi: Conditional score-based diffusion models for probabilistic time series imputation." Advances in Neural Information Processing Systems 34 (2021): 24804-24816.

[8] Kashif Rasul, Calvin Seward, Ingmar Schuster, and Roland Vollgraf. Autoregressive denoising diffusion models for multivariate probabilistic time series forecasting. In Proceedings of the 38th International Conference on Machine Learning, 2021a.

[9] Kashif Rasul, Abdul-Saboor Sheikh, Ingmar Schuster, Urs M. Bergmann, and Roland Vollgraf. Multivariate probabilistic time series forecasting via conditioned normalizing flows. In International Conference on Learning Representations, 2021b.

[10] Biloš, Marin, et al. "Modeling temporal data as continuous functions with process diffusion.” ICML 2023.

---

> ### Author Response · Authors · 2023-12-12
> **Reply to Reviewer Gk9X (Part 1)**
>
> ## Confusing Notations and Setup
>
> $\boldsymbol{x}\_t$ contains the auxiliary information, such as day of the week at time $t$ (encoded as a vector); see Section 1.3 (Paragraph 3) and Section 3 (Paragraph 1, after Assumption 1). Meanwhile, the prediction of $\boldsymbol{y}\_t \in R^{D\_y}$ depends on $\boldsymbol{x}$ as well as the historical observations of *all* dimensions of the multivariate time series itself, $\boldsymbol{y}\_{t-B:t-1}$. The prediction of each dimension, $y\_{t,i}$, depends on both the past observations of itself, $y\_{t-B:t-1,i}$, and of all the other dimensions, $y\_{t-B:t-1,j}, j \ne i$.
>
> ## Multivariate vs Univariate
>
> Indeed we make a diagonal covariance structure assumption for the target $\boldsymbol{y}\_t$ conditioned on the history of *all* dimensions of the target $\boldsymbol{y}$ and the auxiliary variable $\boldsymbol{x}$. We highlighted this assumption in the abstract and paper (as the reviewer noted) to avoid any confusion. Because of this assumption, we do not focus on learning the correlations between the output dimensions of $\boldsymbol{y}\_t$. However, we disagree that our model should therefore be called "univariate", because (1) our model still considers the *lagged* covariance between different dimensions due to conditioning on the history of all dimensions; (2) univariate time series models usually refer to models that use the history of $y\_{<t,i}$ itself to predict $y\_{t,i}$ for each dimension $i$, without considering even the *lagged* covariance between different dimensions; (3) in the literature, multivariate time series forecasting models that assume conditional independence in the output dimensions do exist as well:
>
> - Point forecasting approaches, such as [2,4], do not model output covariance at all but are still considered multivariate, as long as the input and output consist of all dimensions of $\boldsymbol{y}$.
> - Some multivariate probabilistic forecasting models, such as "Vec-LSTM ind-scaling" used as baselines in [8,9,11], also make the same conditional independence assumption as we do.
>
> It is also worth mentioning that the conditional independence assumption, although limiting flexibility, can act as an implicit regularizer and help the model avoid learning spurious correlations between the output dimensions.
>
> ## CRPS-Sum
>
> **CRPS-Sum**, proposed in [11] as an alternative metric to the standard CPRS [12], is defined as the CRPS of the *sum* of all dimensions of $\boldsymbol{y}\_t$, i.e., CRPS of $\sum\_i y\_{t,i}$, while **CRPS** in the multivariate setting (as reported by us and in the literature) is the average CRPS across all dimensions. The summation of $y\_{t,i}$ in CRPS-Sum can be problematic, because errors in different dimensions can cancel each other out, so even if the model made worse predictions for individual dimensions, which would lead to worse CRPS, it may still achieve better CRPS-Sum. This may lead to inconsistencies between CRPS-Sum and other common metrics, which can be found in the previous published results. Therefore, we focus on the other two most commonly used metrics, CRPS and MSE, for our evaluation.

---

> > ### Author Response · Authors · 2023-12-12
> > **Reply to Reviewer Gk9X (Part 2)**
> >
> > ## Missing Citations and Baselines
> >
> > We thank the reviewer for pointing out additional references.
> >
> > Regarding the references related to non-stationary time series modeling, we want to highlight the difference between non-stationary *marginal* vs *conditional* distributions, which we discussed in Section 2 (related work) of our paper. In [1], for example, the authors propose non-stationary temporal matrix factorization, but the non-stationarity is defined and addressed as seasonality and trend (see Section 1 and 3.2 in [1]), and their approach is to build the differencing operator into the optimization itself (see Equation 3 and 4 in [1]). These are discussed in our paper under **data transformation** in related work for handling marginal non-stationarity, but it is not the focus of our work. We deal with this type of non-stationarity by adding additional information, such as day of the week, similar to many previous methods, such as [8,9,11].
> >
> > [6,7] developed diffusion models for time series imputation, which can be used for forecasting if the observations at future time steps are treated as missing. However, for forecasting, the ideas are similar to TimeGrad [8], in which the denoising function $\epsilon\_\theta$ also conditions on the past observations.
> >
> > We added comparisons to 4 more baselines [2,4,5,10] and citations/discussions in the related work (Section 2) to the others [1,3,6,7].
> >
> > ## Qualitative Evaluation
> >
> > We performed a qualitative evaluation on synthetic datasets (Figure 3), because the non-stationarity in the conditional distribution depends on the true distribution of the data, which we do not have access to for real-world datasets. In Figure 3, we can see how our model reacts and adapts to different types of changes (discrete and continuous, seen and unseen) in the conditional distribution. In the revision, we also added new forecasting plots (Appendix G, Figure 4) comparing DynaConF with StatiConF on a specific instance. However, we note that since we do not have access to the true data distribution, the forecasting plots serve more as an illustration than qualitative evaluation in contrast to Figure 3.
> >
> > ## Unrolling of the RNN
> >
> > In our model, the RNN (or any other encoder architecture choice) takes the past observations $\boldsymbol{y}\_{t-B:t-1}$ (omitting auxiliary information $\boldsymbol{x}\_{t-B:t}$ here for simplicity) and encodes them into a fixed-dimensional vector $\boldsymbol{h}\_t$ (its last hidden state), which then determines the predicted distribution of $\boldsymbol{y}\_t$ conditioned on $\boldsymbol{y}\_{t-B:t-1}$. Assume the forecast step size is 1, and now we need to predict $\boldsymbol{y}\_{t+1}$. The RNN would unroll from the initial hidden state again but take $\boldsymbol{y}\_{t-B+1:t}$ as the input sequence to generate the hidden state $\boldsymbol{h}\_{t+1}$, instead of using the final hidden state of the previously unrolled RNN, $\boldsymbol{h}\_t$, and only unrolling for an additional step with $\boldsymbol{y}\_t$, as is typically the case in the existing work. In our case, the RNN encodes $\boldsymbol{y}\_{t-B:t}$ in the same way for any $t$, as it always unrolls from scratch, while in the existing work, how/what it encodes would depend on the length of the past observations it has unrolled on before $t$, which would change according to $t$.

---

> > > ### Author Response · Authors · 2023-12-12
> > > **Reply to Reviewer Gk9X (Part 3)**
> > >
> > > ## References
> > >
> > > [1] Chen, Xinyu, et al. "Nonstationary temporal matrix factorization for multivariate time series forecasting." arXiv preprint arXiv:2203.10651 (2022).
> > >
> > > [2] Liu, Yong et al. “Non-stationary Transformers: Exploring the Stationarity in Time Series Forecasting.” Neural Information Processing Systems (2022).
> > >
> > > [3] Yanchenko, Anna K., and Sayan Mukherjee. "Stanza: a nonlinear state space model for probabilistic inference in non-stationary time series." arXiv preprint arXiv:2006.06553 (2020).
> > >
> > > [4] Woo, Gerald, et al. "Deeptime: Deep time-index meta-learning for non-stationary time-series forecasting." arXiv preprint arXiv:2207.06046 (2022).
> > >
> > > [5] Drouin, Alexandre, Étienne Marcotte, and Nicolas Chapados. "Tactis: Transformer-attentional copulas for time series." International Conference on Machine Learning. PMLR, 2022.
> > >
> > > [6] Lopez Alcaraz, Juan Miguel, and Nils Strodthoff. "Diffusion-based time series imputation and forecasting with structured atate apace models." Transactions on machine learning research (2023): 1-36.
> > >
> > > [7] Tashiro, Yusuke, et al. "Csdi: Conditional score-based diffusion models for probabilistic time series imputation." Advances in Neural Information Processing Systems 34 (2021): 24804-24816.
> > >
> > > [8] Kashif Rasul, Calvin Seward, Ingmar Schuster, and Roland Vollgraf. Autoregressive denoising diffusion models for multivariate probabilistic time series forecasting. In Proceedings of the 38th International Conference on Machine Learning, 2021a.
> > >
> > > [9] Kashif Rasul, Abdul-Saboor Sheikh, Ingmar Schuster, Urs M. Bergmann, and Roland Vollgraf. Multivariate probabilistic time series forecasting via conditioned normalizing flows. In International Conference on Learning Representations, 2021b.
> > >
> > > [10] Liu, Yong, Chenyu Li, Jianmin Wang, and Mingsheng Long. "Koopa: Learning Non-Stationary Time Series Dynamics with Koopman Predictors." Neural Information Processing Systems (2023).
> > >
> > > [11] Salinas, David, Michael Bohlke-Schneider, Laurent Callot, Roberto Medico, and Jan Gasthaus. "High-Dimensional Multivariate Forecasting with Low-Rank Gaussian Copula Processes." In Advances in Neural Information Processing Systems, 6824–34, 2019.
> > >
> > > [12] Matheson, James E., and Robert L. Winkler. "Scoring Rules for Continuous Probability Distributions." Management Science 22, no. 10 (1976): 1087–96.

---

### Review · Reviewer_Zztg · 2023-11-30

**Summary Of Contributions:**

The paper proposes DynaConF to tackle the non-stationary conditional distributions in time series forecasting. DynaConF employs a decomposable modeling architecture to capture the time-variant and time-invariant parts separately. Concretely, these two parts are designed as fixed model parameters and changed distributions respectively. Experimentally, DynaConF performs best in various time series tasks.

**Audience:**

Yes

**Broader Impact Concerns:**

This paper only focuses on technical designs. Thus, there are no ethical risks. Since this paper focuses on the time series forecasting, the proposed method can be used in taffic planning and weather forecasting.

**Claims And Evidence:**

Yes

**Requested Changes:**

In general, I think this paper is well-qualified. Ablations and comparisons to more baselines are expected.

**Strengths And Weaknesses:**

### Strengths

-	This paper is well-written.

-	The idea of decomposable modeling for non-stationary time series is interesting. The overall design of DynaConF is convincing and reasonable.

-	Extensive experiments are included.

### Weaknesses
-	The ablation studies for the decomposition of time-variant and time-invariant parts are not missing, which is important to verify the effectiveness of each design in DynaConF.

-	Some relevant baselines are not compared and discussed, such as Koopa [1] which also employs a variant-invariant decomposition.

[1]  Koopa: Learning Non-stationary Time Series Dynamics with Koopman Predictors, NeurIPS 2023

- There are many deterministic forecasting models, such as Autoformer (NeurIPS 2021), PatchTST (ICLR 2023). The author should also consider to compare DynaConF with them. For example, they can adopt them to predict both mean and deviation simultaneously.

---

### Author Response · Authors · 2023-12-12

We thank all reviewers for their thoughtful comments. We have updated the paper with major changes highlighted, including the addition of the following 4 baselines based on the suggestions from reviewer Gk9X and Zztg, and citations/discussions for the other related work pointed out by the reviewers.

[1] Liu, Yong et al. “Non-stationary Transformers: Exploring the Stationarity in Time Series Forecasting.” Neural Information Processing Systems (2022).

[2] Woo, Gerald, et al. "Deeptime: Deep time-index meta-learning for non-stationary time-series forecasting." arXiv preprint arXiv:2207.06046 (2022).

[3] Drouin, Alexandre, Étienne Marcotte, and Nicolas Chapados. "Tactis: Transformer-attentional copulas for time series." International Conference on Machine Learning. PMLR, 2022.

[4] Liu, Yong, Chenyu Li, Jianmin Wang, and Mingsheng Long. "Koopa: Learning Non-Stationary Time Series Dynamics with Koopman Predictors." Neural Information Processing Systems (2023).

---

### Decision · Action_Editor_iLem · 2024-01-19

**Recommendation:** Accept as is

**Comment:**

Reviewers agree that the paper is well-written and includes an extensive evaluation to support the authors' claims. The main concerns were regarding the novelty of the method and the lack of state-of-the-art results on some of the tasks. Nevertheless, the paper includes a broad survey of related work and, after the revision, comparisons with many relevant baselines. This allows the reader to assess the pros and cons of each respective method in light of their performance on the chosen benchmark tasks.

**Audience:**

Yes.

**Claims And Evidence:**

Yes. The authors have responded well to challenges from reviewers regarding their claims and submitted a revision in which their claims were clarified further.